# RASRAG: A Domain-Specific RAG Framework and Benchmark for Robotic-Assisted Surgery

## Abstract

Robot-assisted surgery (RAS) has significantly improved patient outcomes by reducing blood loss, shortening hospital stays, and accelerating recovery. Despite these benefits, the widespread adoption of RAS has been slowed by a shortage of trained robotic surgeons and limited access to robotic systems. One of the major limitations is access to academic materials and expertise in this domain, which are mostly limited to private company programs or a few textbooks. In this regard, foundation and large language models (LLMs) have been shown to excel in information retrieval and knowledge synthesis. However, none have been specifically adapted to the complexities of the RAS domain. To address this gap, we introduce RASRAG, a RankLLaMA-based Tree Retrieval-Augmented Generation framework that leverages a hierarchical structure derived from the source textbook. Our contributions are: (1) a novel tree-based RAG architecture in which RankLLaMA jointly performs agentic exploration and reranking along the hierarchy ("forest of knowledge"), yielding more relevant retrieval than embedding-only baselines, fine-tuned models, and alternative RAG methods; (2) a publicly available, first-of-its-kind question–answer benchmark curated by five surgeons and two physicians, reflecting real-world RAS clinical inquiries; and (3) clinically grounded evaluation protocol, including blind grading of both model and human answers by surgeons and RAG-specific retrieval and answer quality measures. RASRAG with significantly smaller models matches or outperforms state-of-the-art LLMs, fine-tuned LLMs, and existing RAG architectures regarding precision and relevance for domain-specific tasks.

## 1 Introduction

Robotic-assisted surgery (RAS) has emerged as a preferred platform for delivering minimally invasive surgery (MIS), owing to its enhanced dexterity, superior visualization, and ergonomic benefits Fong et al. (2025). However, despite its increasing use, the adoption of RAS remains uneven. Many surgeons continue to face substantial barriers to obtaining the necessary training, and the specialized skills required for MIS are not evenly distributed across the surgical workforce Cole et al. (2018). Consequently, access to RAS, and in particular to proficient surgeons and institutional infrastructure, remains limited in many areas, contributing to disparities in care and the persistence of the so-called MIS deserts Schneider et al. (2021).

Large Language Models (LLMs) have recently shown great promise in synthesizing large amounts of information and providing high-quality contextual insights Bommasani et al. (2021). For example, OpenAI's GPT-4 OpenAI et al. (2024) has demonstrated innovative capabilities across various applications. These models have been widely adopted in fields such as education Antu et al. (2023); Li et al. (2023b) and medicine Singhal et al. (2025); Sallam (2023); Thirunavukarasu et al. (2023). LLaVA-Med Li et al. (2023a), for instance, integrates vision and language models to interpret medical images through natural language prompts. Similarly, BioGPT Luo et al. (2022) and MedPaLM 2 Qian et al. (2024) are specialized in generating accurate responses to clinical questions. LLMs are also beginning to show promise in surgical contexts, including scientific writing Altmäe et al. (2023), diagnostic imaging Liu et al. (2024), and preoperative management Cheng et al. (2023).

GPT-4 was recently evaluated on a two-part surgical board examination, achieving accuracy rates between $63.6\%$ and $83.3\%$ across various specialties Oh et al. (2023). Additionally, SurgeryLLM Ong

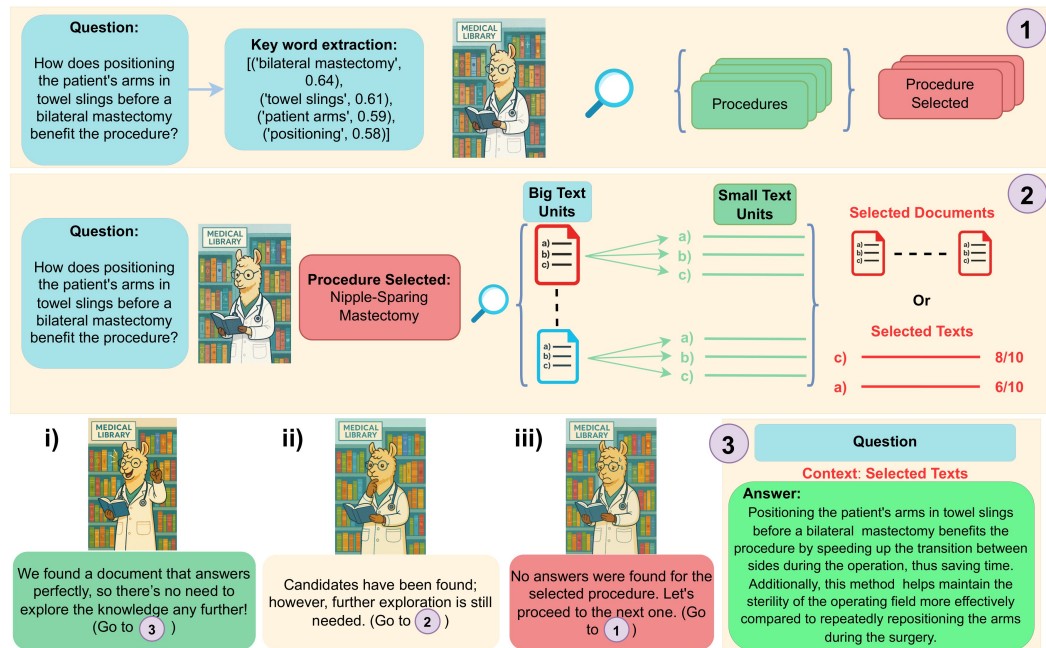

Figure 1: High-level schematic of the RASRAG search process, designed to mimic an expert's workflow for finding information in a surgical textbook. (1) RASRAG begins by extracting keywords from the user query to identify the most relevant surgical procedure. (2) Once a procedure is selected, the system performs a hierarchical search through its associated documents, (ii) collecting a set of potential candidate contexts along the way. (i) If a definite context that fully answers the query is found, the search terminates immediately, and that context is passed to the LLM for answer generation. (iii) However, if the search of the procedure's documents completes without finding any relevant context, the system moves to the next most promising procedure and repeats the process.

et al. (2024) employed a Retrieval-Augmented Generation (RAG) framework to incorporate clinical knowledge and generate patient-specific surgical recommendations. However, no language model has been designed specifically for RAS. Existing models Wu et al. (2024), Li et al. (2023a), and Luo et al. (2022) incorporate surgical data only within broader training corpora He et al. (2025); Pal et al. (2022), with no documented efforts to train explicitly on RAS knowledge.

Retrieval-augmented generation (RAG) systems combine LLMs with external knowledge sources to improve factual accuracy Lewis et al. (2021). In practice, a RAG pipeline first retrieves the top-$k$ relevant passages using dense vector retrievers such as DPR Karpukhin et al. (2020); Lewis et al. (2021). These candidate passages are then optionally reranked by neural models to prioritize the most useful information Ma et al. (2024). Such techniques have been widely applied in medical question-answering. Using domain-specific corpora (e.g., PubMed abstracts, clinical notes), RAG reduces hallucinations and improves reliability in medical QA Ngo et al. (2024). However, simple top-$k$ retrieval may struggle with complex queries that require reasoning over a hierarchy of concepts. To address this, recent methods build structured retrieval paths. For example, graph-based systems link extracted entities into medical knowledge graphs Wu & *et al.* (2024), and multi-agent RAG frameworks coordinate specialized retrievers across multiple sources. Others exploit LLM "planning" or "tree-of-thought" strategies. For example, Fatehkia et al. (2024) uses a tree of entity contexts to augment RAG, while Li et al. (2024) uses the Tree-of-Reviews to explore or prune branches during multi-hop retrieval dynamically.

This study presents a high-precision RAG framework for robotic-assisted surgery, built on a semantic tree representation of the leading published textbook Giulianotti et al. (2023) and powered by the RankLLaMA model Ma et al. (2024) (Figure 1). At each node of the tree, a fine-tuned LLaMA-2 reranker compares the user's query with candidate text chunks from child nodes, selecting the most relevant path and discarding others. Unlike previous tree-based retrieval approaches Li et al. (2024), our method uses reranking not to expand reasoning, but to guide semantic navigation through a

structured corpus. This dynamic path selection improves retrieval precision, avoiding irrelevant sections and yielding accurate, context-sensitive answers from our curated database, often with visual support. Designed to assist surgeons, residents, and educators, this system also lays the foundation for future developments in autonomous robotic-assisted surgery.

Robotic-assisted surgery (RAS) presents a unique scenario: the knowledge base is sparse, highly specialized, and semantically siloed. Only a few shared sections share semantics, such as the instrument requirements or the trocar placement, and the details are crucial for distinguishing procedures that appear similar. For example, robotic cholecystectomy instrument requirements largely overlap with those of a robotic low anterior resection, but differ in fine-grained details. For this reason, we prioritize high precision and domain-specificity over generality. Minor factual inaccuracies (e.g., specifying the wrong endoscope angle, $0°$ instead of $30°$) can invalidate an otherwise correct instrument list (in this case, by compromising depth perception, a critical factor in RAS).

Therefore, our framework is guided by two primary design goals: (i) Traceability, ensuring that all answers are grounded in validated sources so users can verify the provenance of the information; and (ii) Deployment feasibility, as our system is optimized for clinical and academic settings where massive models like GPT-5 may be impractical due to the lack of transparency. Given these requirements and current studies Ovadia et al. (2023); Gekhman et al. (2024), we focus on a retrieval architecture to ensure the highest possible precision, and not on fine-tuning models as they have proved to increase hallucination rates, and could also lead to model misalignment Betley et al. (2025). An example of a question-answer can be found in Figure 2.

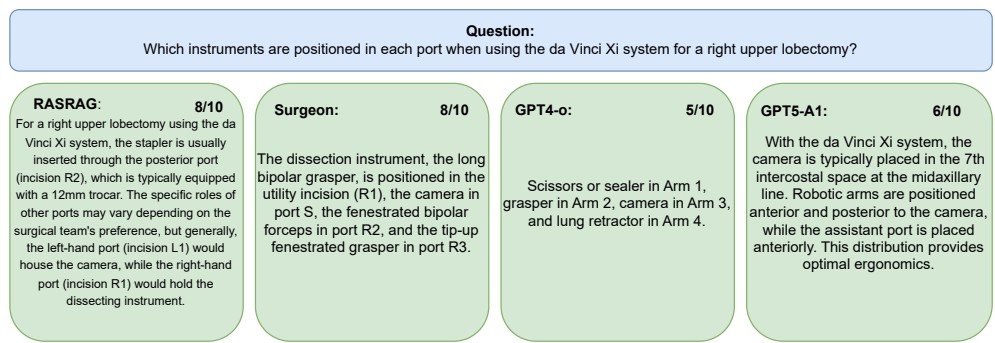

Figure 2: Question Answer example graded by 3 surgeons that demonstrates RASRAG's clinical precision. Our system retrieves and contextualizes current, evidence-based standards for RAS. This output contrasts with that of baseline models, which may offer an incorrect, or imprecise answer. This example underscores our system's commitment to clinical detail and highlights the need for evaluation methods that can recognize and reward such nuance.

## 2 RASRAG: EXPLORING TREE-DATABASES WHILE DOING RERANKING

### 2.1 ANSWERING RAS QUERIES

RAS lacks the comprehensive textbooks and standardized curricula available for traditional surgical disciplines, instead relying on a patchwork of vendor-led modules, isolated academic papers, and institutional training of varying quality (e.g., Intuitive Surgical (2025)). Often, the training primarily addresses technical operation rather than clinical decision-making. Although structured training models emphasize core components such as didactics and console practice, these programs remain largely institution-specific and unstandardized. Furthermore, robotic surgery requires new theoretical knowledge, such as optimal endoscopic or trocar positions to minimize arm collisions. To bridge this gap, new textbooks are emerging, such as Costello (2023) and Giulianotti et al. (2023). This study uses Giulianotti et al. (2023) to design a knowledge database as a knowledge tree. The textbook describes 30 RAS procedures, explaining the preoperative setup (instrument requirements, patient placement, trocar placement), contraindications, and surgical steps in each case while highlighting possible anatomical variations. Therefore, the knowledge forest is composed of 30 trees, one for each procedure (chapter of the textbook).

We created two classes of non-overlapping text units using the RecursiveCharacterTextSplitter Chase (2022). **Small-Text Units (STUs)** use a `chunk_size` of 1,000, while **Big-Text Units (BTUs)** use a `chunk_size` of 10,000. The distribution of the resulting STUs for each procedure is shown in Figure 17. This process also supports a multimodal output; by storing all textbook images, any figure referenced within a retrieved text unit can be loaded and presented to the user. Since this feature is implemented via straightforward string matching between the text and image names, we do not elaborate on it further in this study.

## 2.2 STATISTICAL ASSUMPTIONS

To keep our retrieval search tractable while ensuring completeness within each unit (e.g., avoiding the retrieval of partially explanatory chunks), we make three hierarchical assumptions. The evidence we used to justify these assumptions can be found in Appendix A.1.

**Chapter-level conditional independence:** We assume that once we select the most promising chapter $Ch_k$, the remaining chapters provide no additional information about the answer $A$.

$$P\big(A \mid Ch_0, Ch_1, \ldots, Ch_n\big) = P\big(A \mid Ch_k\big)$$

This justifies our strategy of exploring chapters sequentially and stopping the search as soon as a sufficiently high-scoring candidate is found (case 4 in Fig. 1).

**BTU/STU insufficiency** Let $\{B_i\}_{i \in I}$ be the BTUs of a chapter, and for each $i \in I$ let $\{S_{i,j}\}_{j \in J_i}$ be its STUs. Let $\mathcal{A}$ be a fixed family of admissible answers.

- Across BTUs: For every proper subset $T \subsetneq I$, there exists $a \in \mathcal{A}$ that is retrieved by $\{B_i : i \in I\}$ but not retrieved by $\{B_i : i \in T\}$.
- Within a BTU: For each $i \in I$ and every proper subset $U \subsetneq J_i$, there exists $a \in \mathcal{A}$ that is supported by $\{B_k : k \in I \setminus \{i\}\} \cup \{S_{i,j} : j \in J_i\}$ but not supported by
  $\{B_k : k \in I \setminus \{i\}\} \cup \{S_{i,j} : j \in U\}$.

Any proper subcollection of BTUs (or STUs within a chosen BTU) can miss some answers; hence the search must (i) locate a relevant BTU and (ii) scan *all* of its STUs.

## 2.3 SEARCH STRATEGY

To improve upon traditional cosine similarity methods, which often suffer from inductive bias and are heavily dependent on the quality of the embedding space, we adopt an LLM-based search strategy. At the core of this approach is RankLLaMA Ma et al. (2024), a LLM-based ranking engine that evaluates the semantic relevance between a query and a set of candidate texts. Unlike embedding-only techniques, RankLLaMA leverages full language understanding, allowing for more accurate scoring even when lexical overlap is low or when the query involves nuanced intent.

We model the decision process as a single tree, $T$, represented by a nested dictionary structure. The tree initially branches into 30 main nodes, $\{r_i\}_{i=1}^{30}$, each corresponding to a specific RAS procedure. Each procedure $r_i$ is composed of $m_i$ BTU nodes, $\{b_{ij}\}_{j=1}^{m_i}$. Finally, each BTU node $b_{ij}$ branches into $n_{ij}$ STU leaf nodes, $\{s_{ijk}\}_{k=1}^{n_{ij}}$, which represent subordinate explanatory elements or sub-decisions. Thus, the tree can be formally described as a mapping:

$$T : \Big\{r_i \mapsto \big\{b_{ij} \mapsto \{s_{ijk}\}_{k=1}^{n_{ij}}\big\}_{j=1}^{m_i}\Big\}_{i=1}^{30} \tag{1}$$

Given a user query $q$, our objective is to identify the most relevant root node $r^*$ from the decision tree $T$. The procedure begins by extracting the top five keywords from the query using KeyBERT: $kw = \mathrm{KeyBERT}(q)$. Next, each root node $r_i$ is ranked against the keywords using the RankLLaMA scoring function, $\mathrm{rank}(\cdot, \cdot)$, which evaluates the relevance between its two inputs. We then select the two highest-scoring root nodes:

$$i_1^* = \arg\max_i \{\mathrm{rank}(kw, r_i)\}, \quad i_2^* = \arg\max_{i \neq i_1} \{\mathrm{rank}(kw, r_i)\} \tag{2}$$

For each candidate root node $r_{i*}^*$, we compute the average scores of the BTUs against the query $q$, $\mu_{i^*} = \frac{1}{m_{i^*}} \sum_{j=1}^{m_{i^*}} \text{rank}(q, b_{i^*j})$. We then select the candidate with the highest $\mu_{i^*}$:

$$r^* = r_{x^*}^*, \quad x^* = \arg \max_{x \in \{i_1^*, i_2^*\}} \mu_x \tag{3}$$

We define a set of relevance thresholds:

$$\left\{ \tau_{\text{def}}^B, \ \tau_{\text{cand}}^B, \ \tau_{\text{def}}^S, \ \tau_{\text{cand}}^S, \ \tau_{\text{uncertain}}^S \right\}$$

along with an exploration cap, $\kappa_{\max}$ (formerly the "stubbornness limit"). These are treated as tunable hyperparameters. The relevance thresholds are tuned by doing sensitivity analysis (Appendix A.2) with illustrative examples provided in Appendix A.5.

From this chosen $r^*$, let $B = \{b_{x^*j}\}_{j=1}^{m_{x^*}}$ and

$$b_{x^*j} \in \begin{cases} B_{def} & \text{if} \quad \text{rank}(q, b_{x^*j}) > \tau_{\text{def}}^B \\ B_{cand} & \text{if} \quad \tau_{\text{def}}^B > \text{rank}(q, b_{x^*j}) > \tau_{\text{cand}}^B \\ \emptyset & \text{otherwise} \end{cases} \tag{4}$$

If the set of definite BTUs ($B_{\text{def}}$) is not empty ($B_{\text{def}} \neq \emptyset$), its elements are returned immediately as precise hits, and the search terminates as a direct application of the Chapter-level conditional independence. Otherwise, the procedure explores the set of STUs ($S = \{s_{x^*y^*k}\}_{k=1}^{n_{x^*y^*}}$) contained within each candidate BTU ($B_{\text{cand}}$), collecting

$$s_{x^*y^*k} \in \begin{cases} S_{def} & \text{if} \quad \text{rank}(q, s_{x^*y^*k}) > \tau_{\text{def}}^S \\ S_{cand} & \text{if} \quad \tau_{\text{def}}^S > \text{rank}(q, s_{x^*y^*k}) > \tau_{\text{cand}}^S \\ S_{uncertain} & \text{if} \quad \tau_{\text{cand}}^S > \text{rank}(q, s_{x^*y^*k}) > \tau_{\text{cand}}^S \\ \emptyset & \text{otherwise} \end{cases} \tag{5}$$

As soon as the set of definite STUs ($S_{\text{def}}$) is not empty, its element is returned, and the process concludes. If it is empty, the set of candidate hits ($S_{\text{cand}}$) is returned instead, after searching through all the STUs. If both $S_{\text{def}}$ and $S_{\text{cand}}$ are empty, the set of uncertain hits ($S_{\text{uncertain}}$) is held as a final fallback option.

If no precise context is found within $r^*$ (i.e., $B_{\text{cand}} = \emptyset$, or $S_{\text{cand}} = \emptyset$), the system initiates a fallback search procedure. This involves iterating through the next most relevant main nodes $r_i$ (by descending $\text{rank}(kw, r_i)$), up to the exploration cap $\kappa_{\max}$. For each main node, the BTU→STU search is repeated until a set of candidate STUs is found ($\text{rank}(q, s) > \tau_{\text{cand}}^S$). Once encountered, its parent main node is selected, and its precise STUs are returned. If the fallback loop completes without finding a precise match, the system returns all previously collected uncertain STUs ($S_{\text{uncertain}}$) as a best-effort answer.

## 3 BENCHMARK DATASET FOR RAS QUERY–ANSWERING (QA)

Due to the limited availability of standardized resources in robotic-assisted surgery, no official benchmark currently exists to evaluate the effectiveness of such systems. To address this gap, we developed a comprehensive benchmark dataset specifically designed to assess LLMs in the context of RAS.

To create the benchmark, a diverse team of seven clinicians (five surgeons and two medical doctors) from varied backgrounds, specialties, and experience levels contributed approximately 10 query-answer (QA) pairs per procedure. This effort resulted in a final dataset of 305 carefully curated QA items, with further details on the distribution shown in Figure 3.

## 4 RESULTS AND DISCUSSION

In this section, we evaluate with three complementary lenses: RAGAS (Es et al. (2024)), NVIDIA Answer Accuracy (Nvidia (2025)), and expert surgeon grading. RAGAS metrics are well-suited to measure retrieval quality, e.g., whether the cited context is precise and the answer is faithful to it, but it can underweight overall answer relevance to the clinical question. NVIDIA Answer Accuracy is a strong evaluator of answer relevance and correctness; however, it is highly dependent on

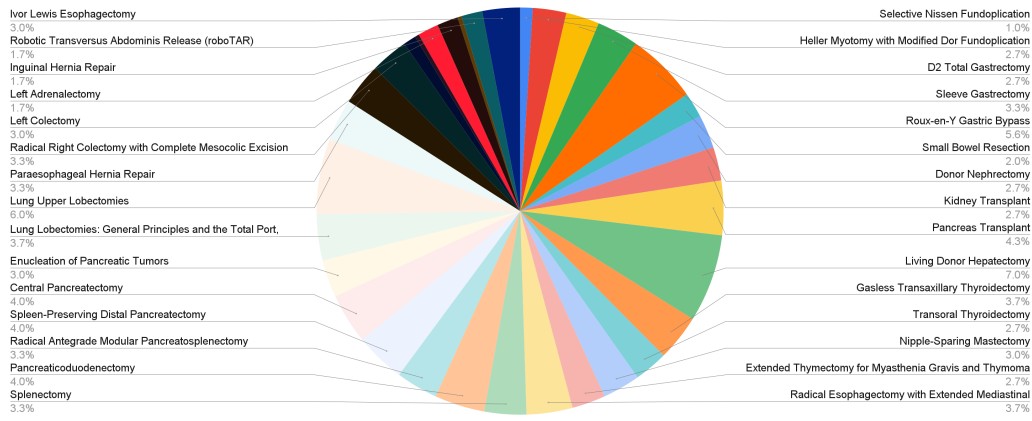

Figure 3: Distribution of benchmark question–answer pairs across robotic-assisted surgeries (30 procedures; 305 questions)

the quality and coverage of the provided ground-truth references—so much so that we also use it to stress-test and refine the benchmark itself. Finally, a key contribution of our study is an expert evaluation by three other independent surgeons, which serves as the domain gold standard for clinical relevance and utility. We report all three in tandem to triangulate performance and interpret divergences between automated metrics and human judgments.

## 4.1 MODEL EVALUATION WITH RAGAS METRICS

Evaluating LLM performance is a significant challenge, complicated by the nuances of natural language that defy traditional machine learning metrics. This difficulty is even more pronounced for RAG systems. Fortunately, dedicated evaluation frameworks are beginning to emerge to address this gap. Recent work has introduced several key benchmarks for RAG, including Saad-Falcon et al. (2023), Es et al. (2024), and Friel et al. (2024), building upon foundational retrieval evaluation studies Gao et al. (2023).

We evaluated our RAG system's performance using the metrics from the RAGAS Es et al. (2024) framework described in Appendix A.3.

Table 1: Evaluation of models using Cosine Similarity and RASRAG

| | Model Info | | Context Precision | Context Recall | Faithfulness | Answer Relevancy | Semantic Similarity | Time (mean) | Time (total) |
|---|---|---|---|---|---|---|---|---|---|
| | Model | Dimension/Size | | | | | | | |
| **Cosine Similarity** | Linq-Embed-Mistral (Kim et al. (2024)) | 4096 | **0.7651** | **0.9092** | 0.8598 | 0.7442 | 0.7713 | 2.0 | 596.5 |
| | multilingual-e5-large-instruct (Wang et al. (2024)) | 1024 | 0.6857 | 0.8219 | 0.8631 | 0.7547 | 0.7714 | 1.8 | 558.1 |
| | jina-embeddings-v3 (Sturua et al. (2024)) | 1024 | 0.6619 | 0.7828 | **0.9167** | 0.6912 | 0.7656 | 5.5 | 1665.4 |
| **Mean** | — | — | 0.7042 | 0.8380 | **0.8799** | 0.7300 | 0.7694 | 3.1 | 940.0 |
| **Std. dev.** | — | — | 0.0540 | 0.0647 | 0.0319 | 0.0340 | 0.0033 | 2.1 | 628.5 |
| **RASRAG** | Llama-3.2-1B-Instruct (Grattafiori et al. (2024)) | 1.24B | 0.8725 | 0.8518 | 0.8388 | 0.7730 | 0.8072 | 12.3 | 3814.4 |
| | Llama-3.2-1B-Instruct_st15 (Grattafiori et al. (2024)) | 1.24B | 0.8674 | 0.8414 | 0.8107 | 0.9016 | 0.7890 | 8.9 | 2774.5 |
| | Qwen2.5-1.5B-Instruct (Team (2024)) | 1.54B | 0.8918 | 0.8579 | 0.7569 | **0.9485** | 0.7793 | 16.2 | 5015.3 |
| | gemma-3-1b-it (Team (2025)) | 1B | 0.8798 | **0.8580** | 0.8845 | 0.7352 | 0.7974 | 15.5 | 4802.1 |
| | Llama-3.2-3B-Instruct (Grattafiori et al. (2024)) | 3.21B | 0.8760 | 0.8554 | 0.8851 | 0.7654 | 0.7983 | 13.3 | 4132.3 |
| | Qwen2.5-3B-Instruct (Team (2024)) | 3.09B | 0.8768 | 0.8555 | 0.8309 | 0.8369 | 0.7699 | 17.9 | 5540.9 |
| | gemma-3-4b-it (Team (2025)) | 4.3B | 0.8794 | 0.8383 | 0.8797 | 0.8090 | 0.7972 | 18.6 | 5755.1 |
| | MedGemma (Sellergren et al. (2025)) | 4B | **0.8829** | 0.8511 | **0.9000** | 0.8228 | **0.8121** | 19.7 | 5992.6 |
| | Mistral-7B-Instruct-v0.3 (Jiang et al. (2023)) | 7.25B | 0.8808 | 0.8573 | 0.8811 | 0.8938 | 0.7853 | 16.9 | 5242.1 |
| | Llama-3.1-8B-Instruct (Grattafiori et al. (2024)) | 8.03B | 0.8835 | 0.8530 | 0.8741 | 0.8548 | 0.7938 | 16.6 | 5140.5 |
| | Qwen2.5-7B-Instruct (Team (2024)) | 7.62B | 0.8778 | 0.8518 | 0.8625 | 0.8780 | 0.7823 | 16.8 | 5214.4 |
| | Gptoss-20B (Agarwal et al. (2025)) | 20B | 0.8760 | 0.8465 | 0.6947 | 0.8627 | 0.7748 | 30.0 | 9113.7 |
| | Qwen2.5-32B-Instruct (Team (2024)) | 32B | 0.8812 | 0.8477 | 0.8187 | 0.8764 | 0.7784 | 32.9 | 10005.4 |
| | Qwen2.5-72B-Instruct (Team (2024)) | 72B | 0.8799 | 0.8494 | 0.7398 | 0.9455 | 0.7787 | 51.8 | 15739.0 |
| | Llama-70B-Instruct (Grattafiori et al. (2024)) | 70B | 0.8805 | 0.8562 | 0.7946 | 0.8990 | 0.7838 | 48.8 | 14820.8 |
| | OpenBio (Ankit Pal (2024)) | 72B | 0.8796 | 0.8529 | 0.8390 | 0.9432 | 0.7918 | 33.3 | 10114.2 |
| | GPT-5 | — | 0.8502 | 0.8202 | 0.7685 | 0.6485 | 0.7631 | 17.5 | 5319.8 |
| | Gemini-2.5-Pro | — | 0.8608 | 0.8334 | 0.8300 | 0.8138 | 0.7766 | 16.9 | 5145.6 |
| **Mean** | — | — | **0.8786** | **0.8520** | 0.8504 | **0.8396** | **0.7900** | 15.3 | 4743.1 |
| **Std. dev.** | — | — | 0.0065 | 0.0069 | 0.0418 | 0.0684 | 0.0110 | 2.9 | 911.6 |

Table 1 presents the performance benchmark of various lightweight models integrated with our RAG framework. We also evaluated several embedding models on a conventional RAG setup, which retrieves the context based on the cosine similarity between the embedded context and input query, and generates responses using Llama-3.2-1B-Instruct Grattafiori et al. (2024). Gemma3-27B Team (2025) was selected as the judge model due to its recent release, strong performance on standard

benchmarks, and suitability for evaluating response quality. The observed performance variance across models highlights the robustness and adaptability of our RAG framework. Notably, the results show that our proposed RASRAG pipeline consistently outperforms the traditional RAG method across all quality dimensions while maintaining strong semantic fluency.

Comparing our RASRAG with the conventional RAG (cosine similarity) using embedding models, the average context precision (0.17 ↑) and average answer relevancy (0.2 ↑) have significantly improved, while the context recall, faithfulness, and semantic similarity have not changed significantly. The low variance among the different retrieval strategies shows that the metrics do not heavily rely on the LLM generating the answers based on the retrieved context. Moreover, it also proves that our retrieval strategy is the key to providing a better context for the model. Llama-3.2-1B-Instruct_st15 is the same model as the one above (Llama-3.2-1B-Instruct), but with reduced stubbornness, i.e. it searches fewer trees when no answers are initially found (cf. 2.1). We observe that by slightly sacrificing faithfulness and context recall, we gain in answer relevancy. These differences primarily confirm that our first statistical assumption (2.2) was valid: once a highly promising candidate is identified early in the ranking, halting the search and returning that result is sufficient. Conversely, if no suitable answer is found among the top-ranked trees, continuing the search yields diminishing returns.

These metrics show that our RAG is not dependent on the model generating the answers (as evidenced by the diverse panel of models we tested 1B models to closed-weight ones), and even with a Qwen2.5-1.5 B-Instruct, it can surpass all cosine-based baselines, highlighting that the retrieval strategy, rather than the model size, is the primary driver of accuracy. While there is a tradeoff between the performance and the average run time, introducing an average 15s latency still remains acceptable for most interactive research and clinical decision-support scenarios.

However, this trade-off is justifiable. In specialized domains like robotic-assisted surgery, relying solely on text embedding similarity is insufficient for accurately retrieving context. A more powerful model, such as RankLLaMA, is necessary to identify truly relevant content. As a result, the additional computation time is warranted to ensure higher-quality retrieval.

The QA evaluation row by row for each model can be found in the attached repository for Table 1.

## 4.2 NVIDIA ANSWER ACCURACY

Table 2: Nvidia metric evaluation on Surgical VQA benchmark.

| Category | Model | Trial 1 | Trial 2 | Trial 3 | Trial 4 | Trial 5 | Mean | Std. Dev. |
|---|---|---|---|---|---|---|---|---|
| **Ground Truth** | N.A. | 0.9762 | 0.9762 | 0.9762 | 0.9762 | 0.9762 | 0.9762 | 0.0000 |
| **Other RAGs** | Linq-Embed-Mistral | 0.6634 | 0.6337 | 0.6328 | 0.6328 | 0.6320 | 0.6389 | 0.0137 |
| | MedGraph+GPT5 | 0.8877 | 0.8877 | 0.8877 | 0.8875 | 0.8875 | 0.8880 | 0.0004 |
| | PaperQA+o4-mini | 0.8270 | 0.8279 | 0.8270 | 0.8270 | 0.8270 | 0.8272 | 0.0004 |
| **Fine-tuned LLMs** | medGemma | 0.3893 | 0.3885 | 0.3885 | 0.3893 | 0.3885 | 0.3888 | 0.0004 |
| | OpenBio | 0.4533 | 0.4525 | 0.4525 | 0.4525 | 0.4525 | 0.4527 | 0.0004 |
| **Proprietary LLMs** | Gemini-2.5-Pro | 0.5828 | 0.5852 | 0.5836 | 0.5844 | 0.5836 | 0.5839 | 0.0009 |
| | GPT-4o | 0.4942 | 0.4983 | 0.4975 | 0.4983 | 0.4975 | 0.4972 | 0.0017 |
| | GPT-5 | 0.5738 | 0.5746 | 0.5730 | 0.5730 | 0.5730 | 0.5735 | 0.0007 |
| **RASRAG** | RASRAG+medGemma | 0.7934 | 0.7943 | 0.7943 | 0.7951 | 0.7934 | 0.7941 | 0.0007 |
| | RASRAG+Mistral-7B | 0.8133 | 0.8127 | 0.8144 | 0.8144 | 0.8127 | 0.8135 | 0.0009 |
| | RASRAG+OpenBio | 0.8221 | 0.8221 | 0.8197 | 0.8213 | 0.8213 | 0.8213 | 0.0010 |
| | RASRAG+Gemini2.5-pro | 0.8303 | 0.8303 | 0.8295 | 0.8303 | 0.8303 | 0.8301 | 0.0004 |
| | RASRAG+GPT5 | 0.8779 | 0.8795 | 0.8779 | 0.8779 | 0.8795 | 0.8785 | 0.0009 |
| **2nd RAS Book Biology Safety Cookbook** | RASRAG+GPT5 | 0.8400 | 0.8400 | 0.8405 | 0.8400 | 0.8400 | 0.8401 | 0.0002 |
| | RASRAG+Mistral-7B | 0.8000 | 0.8050 | 0.8000 | 0.8050 | 0.8050 | 0.8030 | 0.0024 |
| | RASRAG+Mistral-7B | 0.8150 | 0.8150 | 0.8150 | 0.8150 | 0.8150 | 0.8150 | 0.0000 |
| | RASRAG+Mistral-7B | 0.8450 | 0.8450 | 0.8450 | 0.8450 | 0.8450 | 0.8450 | 0.0000 |

The RAGAS evaluation revealed limitations in some standard RAG metrics. For instance, the *faithfulness* score often remained high even when an answer was clinically incongruent with the ground truth, due to an ambiguous process for "claim" extraction. We observed similar issues with *context recall*. Furthermore, metrics like *answer relevancy* can penalize models with superior phrasing capabilities (e.g., GPT-5), as their well-structured answers may deviate in form, though not in substance, from the ground truth, artificially lowering their scores. To overcome these issues and more

directly measure clinical correctness, we adopted the NVIDIA Answer Accuracy Nvidia (2025) metric (Table 2).

To demonstrate the effectiveness of our architecture, we compared the performance of RASRAG against a range of state-of-the-art systems. As shown in Table 1, our implementation significantly outperforms strong baselines. Our comparison included SOTA proprietary LLMs (GPT-4o, GPT-5, Gemini-2.5-Pro), domain-specific fine-tuned models (medGemma, OpenBio), and other specialized RAG architectures (MedGraph, PaperQA). This result supports its strong alignment with expert-level answers in high-precision clinical contexts.

It is important to note key methodological differences for some baseline comparisons. The Med-Graph architecture, for instance, is designed for multiple-choice selection rather than free-form generation; to accommodate this, we created two synthetic incorrect answers to pair with the ground truth, making its task fundamentally simpler than that of our system. Additionally, the PaperQA baseline exhibited significantly higher latency (∼1 min/question), requiring approximately 2-3 times the generation time of RASRAG, despite using a proprietary model not run on local hardware.

To test the generalization capabilities of our RASRAG method, we evaluated its performance on a second, distinct RAS textbook Kim (2014). Because this book covers different procedures and was published a decade prior to Giulianotti et al. (2023), and we generated a new set of 50 question-answer pairs specifically for this evaluation. To demonstrate generalization beyond the primary RAS textbook, we evaluated RASRAG on three additional textbooks: a cookbook (highly structured) National Institutes of Health, Health and Human Services Department (2008) , an invasive plant ecology guide (structured with complex semantics) Huebner & Jones (2022), and a citizen emergency preparedness manual (simpler, with more open-ended QAs) Federal Emergency Management Agency (2013). While this new QA set was not curated by our expert panel and thus may not meet the same quality standard as our primary benchmark, the results are nonetheless informative. Achieving a high NVIDIA Answer Accuracy on this new dataset indicates that RASRAG can effectively retrieve relevant context to support accurate answer generation, even when applied to an entirely different knowledge source.

It is crucial to properly interpret these high-accuracy scores. Consequently, even a very high score, such as the 87% achieved by GPT-5, does not imply that the model surpasses surgeons. Instead, it reflects that its outputs are extremely similar in *content* (not form) to those provided by surgeons, highlighting a shared basis of clinical expertise. This metric is intrinsically unable to demonstrate superiority, as the highest possible outcome is to reproduce answers that are essentially identical in content to those of a human expert.

### 4.3 Evaluation by Experts

To complement the other automated metrics, we conducted a formal evaluation with three independent surgeons who did not participate in the benchmark's creation. They were tasked with grading the answers from our RASRAG+Mistral-7B, GPT-4o, and GPT-5 on a 0-10 scale to assess factual correctness, comprehensiveness, and clinical utility. Further details on the benchmark creation and grading rubric are available in Appendix A.4.

To ensure an unbiased evaluation, all four responses for each question (our system, GPT-4o, GPT-5, and the ground truth) were anonymized and presented to the surgeons in a randomized order. After grading, we measured inter-rater reliability using Krippendorff's alpha Marzi et al. (2024) (95% CI, 1000 bootstrap iterations). The resulting scores, all above 0.67, indicate substantial agreement among the raters and confirm the consistency of the grading process.

The grade distributions, shown in Figure 4, reveal distinct performance patterns (where 0 indicates an incorrect answer, 5 partially correct, and 10 correct). As expected, the ground-truth surgeon responses are clustered at the high end of the scale (8-10). In contrast, GPT-4o and GPT-5 exhibit a much wider distribution with more frequent low-to-mid scores, suggesting lower reliability. Notably, our RASRAG system received the highest concentration of perfect scores (grade 10). RASRAG may generate systematically more comprehensive answers, explicitly stating details that human experts, communicating with peers, often omit due to shared background knowledge. The higher inter-rater agreement when grading RASRAG's outputs supports this interpretation and highlights its consistency and clinical alignment relative to general-purpose proprietary models.

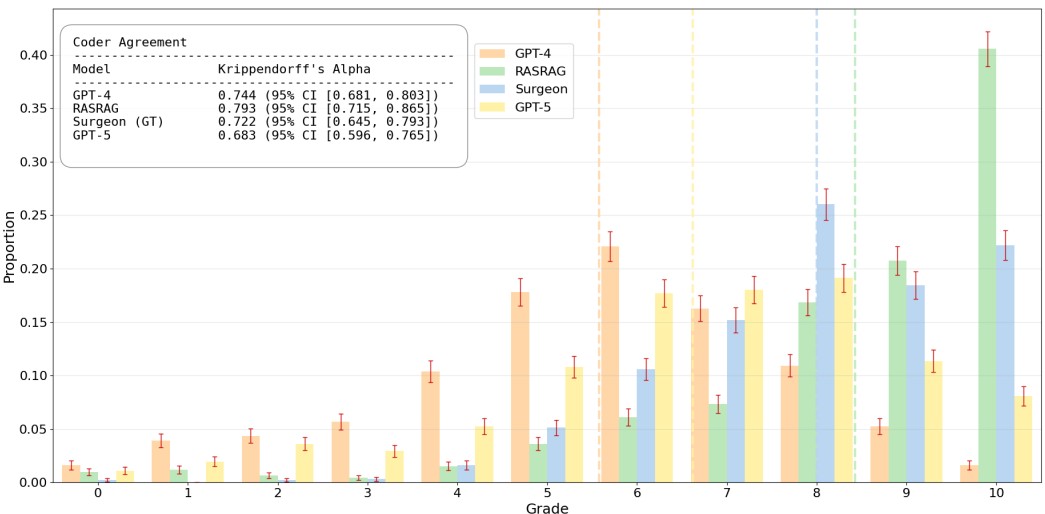

Figure 4: Grades distribution by model: GPT-4o (mean score of 5.58), GPT-5 (mean score of 6.62), RASRAG+Mistral7B (mean score of 8.43), and surgeon's answers (mean score of 8.00). Error bars show the standard error of proportions within each grade category in grading for each model (given 3 graders per question).

## 5 CONCLUSION

This study presents a novel and domain-specific Retrieval-Augmented Generation (RAG) framework tailored for Robotic-Assisted Surgery (RAS) called RASRAG, combining a tree-structured knowledge representation with RankLLaMA-based semantic retrieval. By constructing a hierarchical structure from a leading RAS textbook and applying fine-grained semantic reranking at each node, our RAGAS system enables more context-aware, accurate, and relevant answers compared to traditional embedding-based methods.

Our contributions include: (1) a high-precision RAGAS pipeline powered by RankLLaMA, capable of outperforming cosine-based baselines, scientific specialized models RAG methods, medical RAG, finetuned models in the medical domain and State of the art LLMs, in both quantitative metrics and expert evaluations; and (2) the first publicly available benchmark for RAS QA, curated by five surgeons and two medical doctors, encompassing over 300 high-quality questions and answers that reflect real-world clinical needs. Comprehensive evaluation using RAGAS metrics reveals significant improvements in context precision (approximately $+0.17$), semantic similarity, and answer relevancy, while maintaining high recall and faithfulness. Additionally, a blind evaluation conducted by expert surgeons confirms the strong factual accuracy and clinical usefulness of answers generated by our RankLLaMA + RAGAS pipeline, which closely approached human-written responses and outperformed GPT-4o in quality and consistency. To further confirm this, we validated these results using the Answer Accuracy NVIDIA metric.

We also note that the precision-recall trade-off, along with the increased latency of the semantic reranker, suggests important directions for optimization in real-time clinical applications. These tools will serve as a foundation for scalable, high-accuracy clinical education and decision support. Second, we will explore ways to improve retrieval latency and context coverage by experimenting with hybrid retriever architectures, caching mechanisms, and adaptive thresholding strategies. Finally, we intend to broaden the scope of our QA benchmark by including new procedures, more diverse clinical scenarios, and potentially multimodal data such as annotated surgical videos and intraoperative sensor information.

Altogether, this work represents a step toward closing the knowledge accessibility gap in robotic-assisted surgery and highlights how targeted LLM applications can support medical education, reduce variability in training, and bring us closer to practical AI-powered surgical assistance systems. This methodology could generalize well beyond RAS: for any specialized corpus—such as a newly released scientific book that base LLMs won't absorb promptly—the reasonable choice is to process it through the RASRAG framework, which offers a more robust and timely alternative to fine-tuning.

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

# A APPENDICES

## A.1 SEMANTIC SIMILARITY AND CHAPTER INDEPENDENCE

We computed semantic similarity among BTUs (section-level text units) extracted per chapter using the RASRAG structure. Each BTU was embedded with a SentenceTransformer model (all-mpnet-base-v2), and pairwise cosine similarities were calculated. For intra-chapter analysis, we formed a similarity matrix within each chapter and summarized only the off-diagonal upper-triangle values (self-similarities removed). For inter-chapter analysis, we computed cross-chapter similarity matrices for every chapter pair and flattened all entries. The top histograms plot these intra and inter distributions, the bar chart reports BTU counts per chapter, and the boxplot compares the two distributions directly (intra vs inter).

The analysis reveals a clear separation between intra- and inter-chapter semantic similarities. The top-left histogram shows that intra-chapter BTUs are generally more semantically coherent, with a mean cosine similarity of 0.61 and a distribution skewed toward higher values, indicating stronger contextual alignment within the same chapter. In contrast, the top-right histogram demonstrates that inter-chapter similarities are centered around a lower mean of 0.47, with a symmetric spread and a substantial proportion of weakly related BTUs, reflecting limited semantic overlap across chapters. The bottom-left bar plot confirms that BTUs are unevenly distributed among chapters, which may influence both intra- and inter-chapter similarity distributions. Finally, the bottom-right boxplot reinforces these trends: intra-chapter similarities consistently exceed inter-chapter ones, with a higher median and narrower spread, while inter-chapter similarities show a broader range and multiple high-value outliers, likely representing conceptually related content across chapters. Together, these results support the conclusion that cross-chapter semantic similarity is comparatively low, and intra-chapter content is significantly more cohesive.

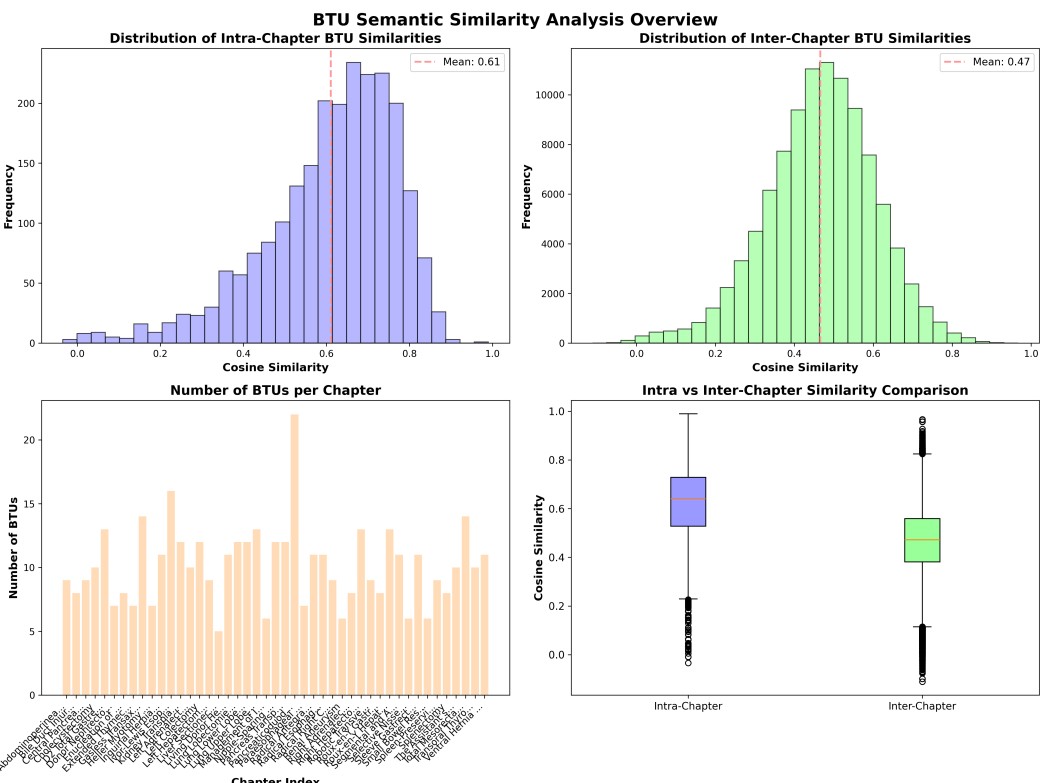

Figure 5: Overview of BTU semantic similarity analysis, showing distributions of intra- and inter-chapter cosine similarities, BTU counts per chapter, and a direct comparison of similarity distributions.

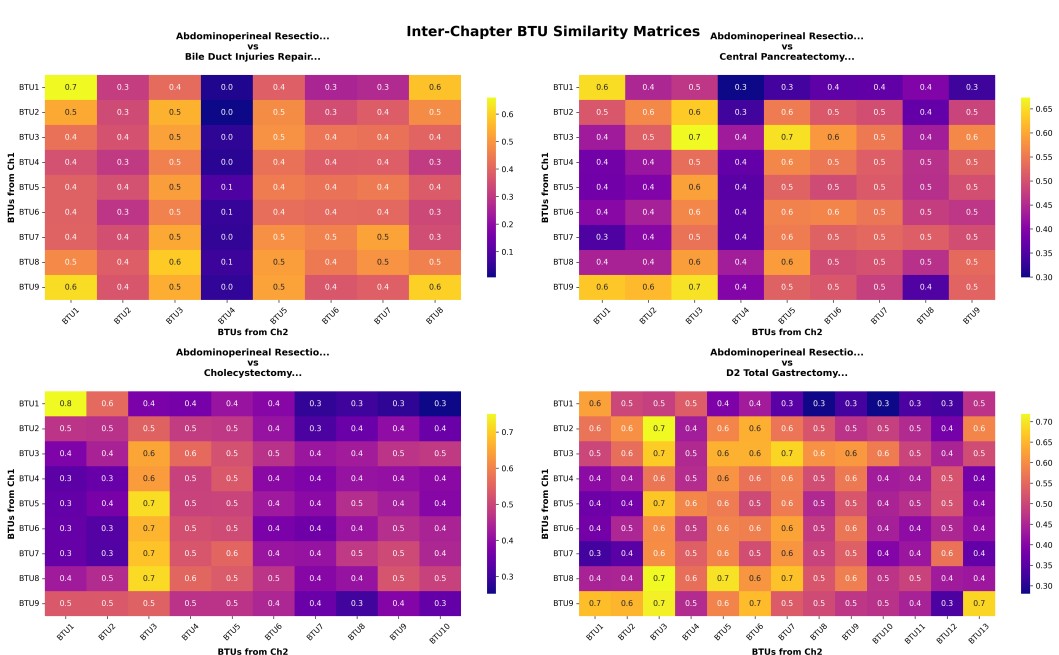

Figure 6: Example of 4 chapters Inter-chapter semantic similarity between BTUs for a given chapter

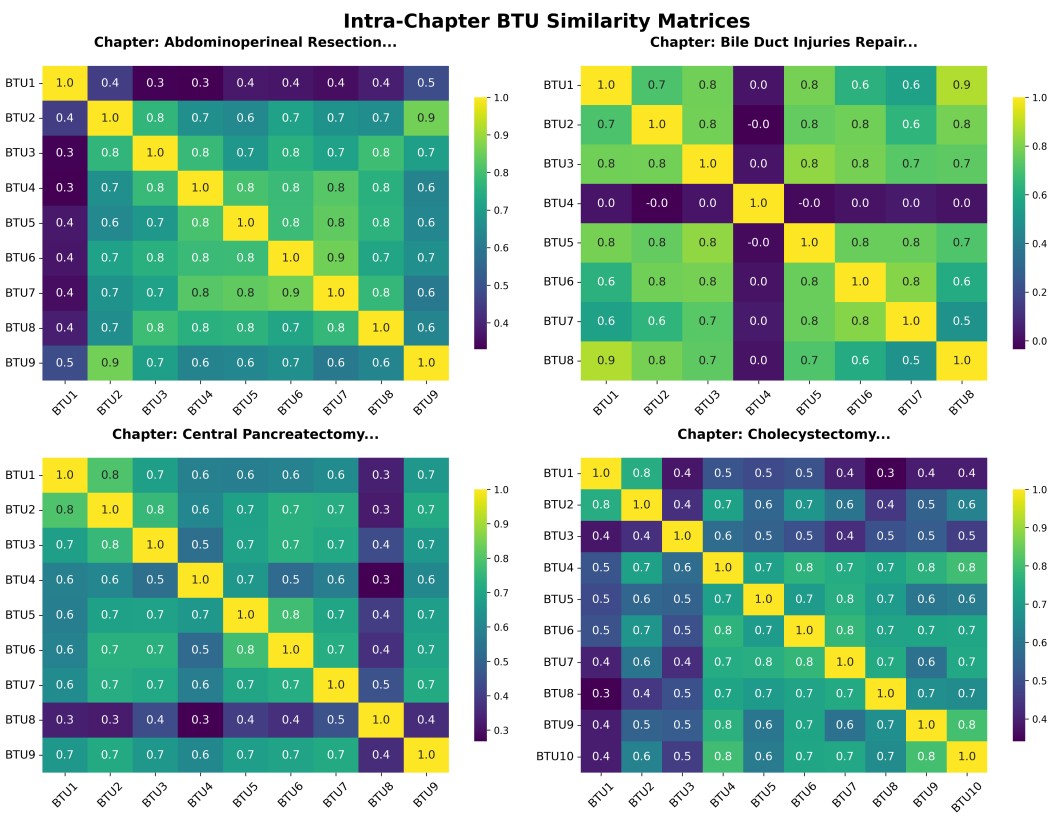

Figure 7: Example of 4 chapters Intra-chapter semantic similarity between BTUs for a given chapter

**Chapter-level relatedness.** Let chapters be indexed by $c$ Chapter $c$ has $n_c$ BTUs, with total $N = \sum_c n_c$. Embed each BTU as $x_i \in \mathbb{R}^d$ and let the centroid of chapter $c'$ be

$$\mu_{c'} = \frac{1}{n_{c'}} \sum_{j \in c'} x_j.$$

For a focal BTU $i \in c$, convert its affinities to all other chapter centroid into a probability distribution using a temperature-controlled softmax, denoted simply as $\mathrm{softmax}(\tau)$:

$$p_{i,\cdot}(\tau) = \mathrm{softmax}(\tau)\big[\, \{\, x_i^\top \mu_{c'} \,:\, c' \neq c \,\} \,\big].$$

Smaller $\tau$ sharpens differences; larger $\tau$ smooths them. If no chapter stands out, $p_{i,\cdot}(\tau)$ is near-uniform. We summarize dispersion with normalized entropy

$$H_{\mathrm{norm}}(i; \tau) = \frac{-\sum_{c' \neq c} p_{i,c'}(\tau) \log p_{i,c'}(\tau)}{\log(C - 1)} \in [0, 1],$$

and optionally track the top mass $p_{\max}(i; \tau) = \max_{c' \neq c} p_{i,c'}(\tau)$.

**Within-chapter cohesion .** Distribute $i$'s similarities over *all* BTUs via $\mathrm{softmax}(\tau)$ and measure the probability mass that returns to its own chapter:

$$q_{i,\cdot}(\tau) = \mathrm{softmax}(\tau)\big[\, \{\, \cos(x_i, x_j) \,:\, j = 1, \ldots, N \,\} \,\big], \qquad \mathtt{own\_mass}(i; \tau) = \sum_{j \in c} q_{i,j}(\tau).$$

A random baseline is $n_c/N$; values well above it indicate cohesive chapters. Very small $\tau$ can cause trivial self-peaks, so we interpret away from saturation.

For these two measure we swept $\tau \in \{0.01, 0.02, 0.03, 0.05, 0.10, 0.20, 0.30, 0.50, 1.00\}$

### A.1.1 RESULTS AND INTERPRETATION.

The $\tau$-sweep in Fig. 8, and more in details Table 3 are reporting the median-of-medians and mean-of-medians for both $H_{\mathrm{norm}}$ and $\mathtt{own\_mass}$, exhibits a clear knee. At very sharp temperatures ($\tau \leq 0.03$), the softmax effectively collapses ($\mathtt{own\_mass} \approx 1$, $H_{\mathrm{norm}} \approx 0.10\text{–}0.48$), rendering the diagnostic uninformative. At $\tau = 0.05$, the distribution remains highly peaked ($\mathtt{own\_mass} \approx 0.84$). By $\tau = 0.10$, we observe modest yet unsaturated within-chapter cohesion ($\mathtt{own\_mass} \approx 0.23$) while the others-only distributions are already highly diffuse ($H_{\mathrm{norm}} \approx 0.93$). For $\tau \geq 0.30$, behavior converges toward uniformity across chapters ($H_{\mathrm{norm}} \geq 0.993$), with $\mathtt{own\_mass}$ receding to the random-allocation baseline $\approx n_c/N$ (e.g., 0.0435 at 0.30, 0.0316 at 0.50, 0.0266 at 1.00). Across all temperatures, $p_{\max}$ remains close to $1/K$ (with $K$ the number of other chapters), indicating that probability mass does not concentrate on any single other chapter. Taken together with the histogram and box-plot separation (intra > inter), these results constitute robust evidence for chapter independence: no systematic inter-chapter pull is present across a broad temperature range, while the only persistent structure is a mild, $\tau$-dependent within-chapter preference.

Figure 8: Example of 4 chapters Intra-chapter semantic similarity between BTUs for a given chapter. Each thin colored line is one chapter; the thick black line is the global median across chapters. These panels plot the normalized entropy $H_{\mathrm{norm}}$ of the others-only distribution and within-chapters ones. Left shows the median across BTUs per chapter; right shows the mean. High values ($\approx 1$) mean probability over other chapters is diffuse (no single other chapter dominates) and thus support inter-chapter independence. Low values indicate a systematic pull to a specific other chapter. (This row is purely inter-chapter.)

Table 3: Temperature sweep (global medians across chapters). $K$ denotes the number of other chapters.

| $\tau$ | $H_{\text{norm}}$ med-of-med | $H_{\text{norm}}$ mean-of-med | own_mass med-of-med | own_mass mean-of-med |
|---|---|---|---|---|
| 0.010 | 0.100 | 0.115 | 1.000 | 1.000 |
| 0.020 | 0.314 | 0.289 | 1.000 | 1.000 |
| 0.030 | 0.484 | 0.464 | 0.992 | 0.991 |
| 0.050 | 0.747 | 0.725 | 0.844 | 0.845 |
| 0.100 | 0.933 | 0.930 | 0.228 | 0.242 |
| 0.200 | 0.984 | 0.984 | 0.066 | 0.069 |
| 0.300 | 0.993 | 0.993 | 0.044 | 0.045 |
| 0.500 | 0.998 | 0.998 | 0.032 | 0.033 |
| 1.000 | 0.999 | 0.999 | 0.027 | 0.027 |

## A.2 SENSITIVITY ANALYSIS

We performed a one-factor-at-a-time sweep over the relevance thresholds to analyze their impact on two key metrics: end-to-end processing time and answer accuracy.

**Impact on Processing Time**    The processing time was measured until an answer was accepted or the fallback loop reached its exploration cap, $\kappa_{\max}$. At the STU level, tightening the thresholds ($\tau_{\mathrm{def}}^{S}$, $\tau_{\mathrm{cand}}^{S}$, $\tau_{\mathrm{uncertain}}^{S}$) monotonically increases processing time, with $\tau_{\mathrm{cand}}^{S}$ showing the steepest growth as it governs the most common early-stopping condition. At the BTU level, raising $\tau_{\mathrm{def}}^{B}$ removes the fast-exit path, sharply increasing mean runtime and variance by forcing deeper STU evaluations. The candidate threshold, $\tau_{\mathrm{cand}}^{B}$, exhibits a U-shaped effect: a low value expands too many BTUs (triggering expensive STU ranking), while a high value frequently causes the candidate set to become empty, activating the computationally intensive root-level fallback loop.

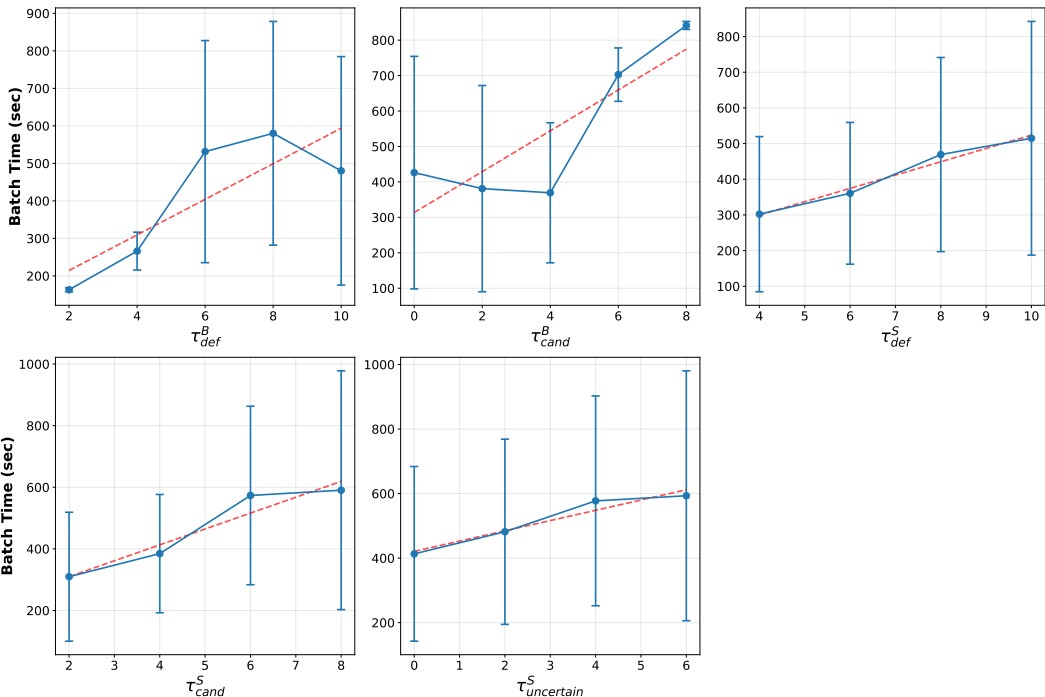

Figure 9: Batch processing time (in seconds) to answer a batch of 60 questions, where markers indicate the mean, error bars represent the standard deviation across hyperparameter variations, and the dashed red line serves as a trend line.

**Impact on Answer Accuracy**    The effects of the thresholds on answer accuracy were also distinct. For the BTU thresholds, accuracy falls sharply as $\tau_{\mathrm{cand}}^{B}$ tightens because the STU search is starved of candidate contexts. Increasing $\tau_{\mathrm{def}}^{B}$ also eventually degrades accuracy, as overly strict gates can discard true positives. The STU thresholds have more subtle effects: the impact of $\tau_{\mathrm{def}}^{S}$ is largely flat, $\tau_{\mathrm{cand}}^{S}$ has a weak concave trend with a mid-range sweet spot, and stricter $\tau_{\mathrm{uncertain}}^{S}$ values monotonically decrease accuracy by reducing recall of borderline-but-useful STUs. At most extreme settings, error bars widen, reflecting an increased reliance on the fallback loop.

**Search Depth Analysis**    Additionally, for fixed search-threshold values, we swept $k_{\max}$ (search depth) from 0 to 30 in steps of 1 and measured the NVIDIA accuracy score. This analysis was performed on 60 randomly sampled questions using RASRAG+Llama-3.2-1B-Instruct. Batch time increases approximately linearly with the deep search parameter, with slope $0.1851$ s per $k_{\max}$ unit, resulting in substantially higher latency for large $k_{\max}$. In contrast, the NVIDIA accuracy exhibits only a very shallow positive trend (slope $0.000798$ per $k_{\max}$ unit), indicating strongly diminishing returns from deeper retrieval.

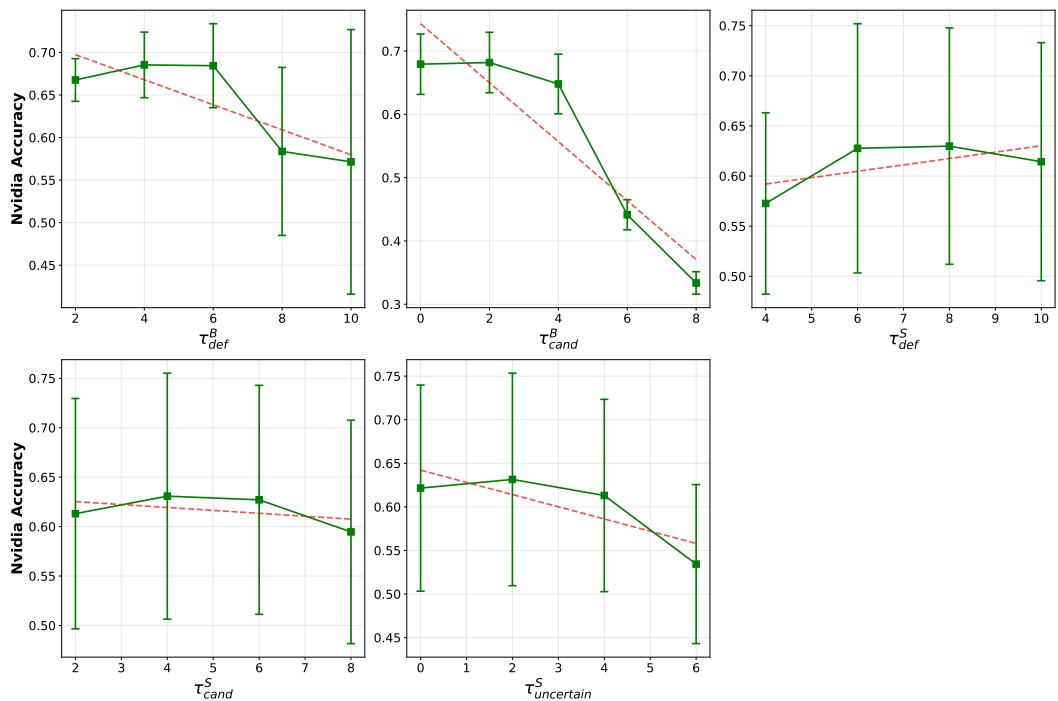

Figure 10: Answer accuracy on a batch of 60 questions, where markers indicate the mean, error bars represent the standard deviation across hyperparameter variations, and the dashed red line serves as a trend line.

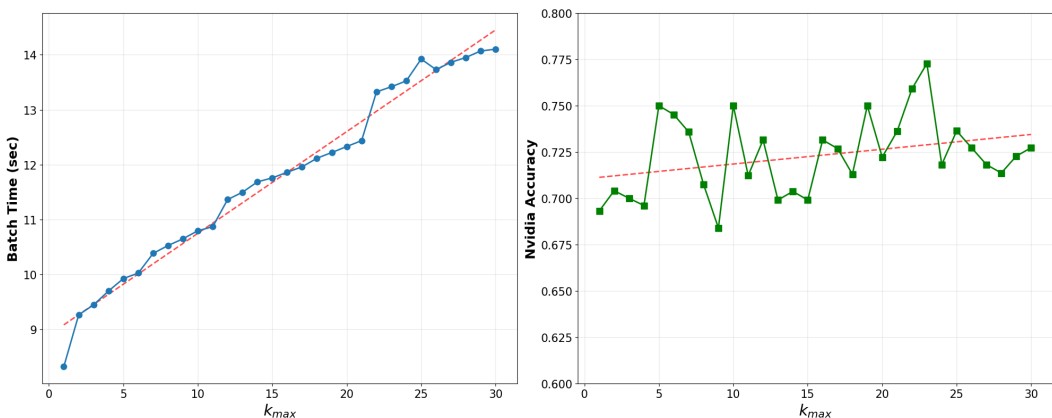

Figure 11: Latency and NVIDIA accuracy as a function of $k_{\max}$ (exploration cap) on a batch of 60 questions. The latency increases with a slope of $0.1851$ s per $k_{\max}$ unit, whereas the NVIDIA accuracy increases only very slightly, with a slope of $0.000798$ per $k_{\max}$ unit.

**Recommended Tuning Strategy**   Taken together, these results inform our final tuning strategy. We recommend a mid-to-low $\tau_{\mathrm{cand}}^{B}$ to balance expansion and fallback risk, a moderate $\tau_{\mathrm{def}}^{B}$ to preserve the fast-exit path, a mid-range $\tau_{\mathrm{cand}}^{S}$ as the primary speed-quality lever, and a permissive $\tau_{\mathrm{uncertain}}^{S}$ to maintain recall. A mid-to-high $\tau_{\mathrm{def}}^{S}$ can be used when answer purity is paramount.

## A.3 RAGAS METRICS

**Context Precision**: Context Precision (CP) quantifies the proportion of relevant chunks within the retrieved contexts. It is computed as the average precision across the top-k ranked chunks. Precision (P) refers to the ratio of relevant chunks among the retrieved items at rank $k$.

$$\text{CP@K} = \frac{\sum_{k=1}^{K} (\text{Precision@k} \times v_k)}{\text{Total number of relevant items in the top } K \text{ results}} \quad (6)$$

$$\text{P@k} = \frac{\text{true positives@k}}{\text{true positives@k} + \text{false positives@k}} \quad (7)$$

Where $K$ is the total number of chunks in retrieved contexts and is the relevance indicator at rank $k$.

**Context Recall**: Assesses the extent to which relevant documents (or information units) have been successfully retrieved. Specifically:

$$CR = \frac{|\text{Number of relevant contexts retrieved}|}{|\text{Total number of reference contexts}|} \quad (8)$$

**Faithfulness**: Evaluates the factual alignment between the response and the retrieved context. This metric also helps identify the amount of noise present in RAG-generated answers. It is formally defined as:

$$F = \frac{|\text{Number of claims in the answer supported by the context}|}{|\text{Total number of claims in the response}|} \quad (9)$$

**Answer Relevancy**: Evaluates how relevant a model's response is to the input query. This metric is calculated by generating a set of artificial questions based on the response. Then compute the cosine similarity between the embedding of the input query ($\mathbf{E}_o$) and the embedding of each generated question ($\mathbf{E}_{g_i}$) and take the average of these cosine similarity scores:

$$\text{Answer Relevancy} = \frac{1}{N} \sum_{i=1}^{N} \text{cosine similarity}(\mathbf{E}_{g_i}, \mathbf{E}_o) \quad (10)$$

where $N$ is the number of generated questions. Higher scores indicate better alignment with the input query, while lower scores are given if the response is incomplete or includes redundant information.

**Semantic Similarity**: Measures how closely the generated response aligns semantically with the ground truth answer. This metric computes the cosine similarity between the embedding of the ground truth answer and the embedding of the generated response. The score ranges from 0 to 1, with higher values indicating greater similarity. This metric provides insight into the quality of the response and is computed using a bi-encoder model that evaluates semantic similarity between the two texts.

## A.4  GRADING SYSTEM

This scoring system adopts a three-level weight structure (60% for correctness, 20% for completeness, and 20% for usefulness), reflecting the progressive dependence and relative independence of these three dimensions. The specific scoring rules are as follows:

1. For correct content (6 points), completeness (1–2 points) and usefulness (1–2 points) will be additionally evaluated, resulting in a total score of 8–10 points.

2. For partially correct content (1–5 points), additional points will be awarded for partial completeness (0–1 points) and usefulness (0–1 points), resulting in a total score of 1–7 points.

3. For incorrect content (0 points), completeness and usefulness will also score 0, resulting in a total score of 0.

In addition, due to the rapid pace of medical advancements, answers involving outdated surgical techniques or the latest innovative technologies will be scored as partially correct at the evaluator's discretion.

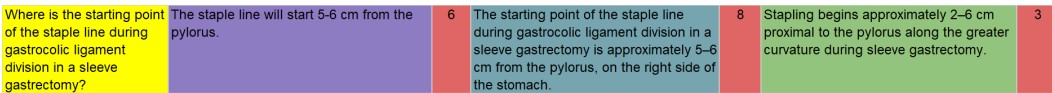

Figure 12: **1rst Answer** : total 6 =4 (partial correct)+1 (partial completeness)+1 (partial completeness), **2nd Answer**: total 8 =6 (correct)+1 (partial completeness)+1 (partial usefulness), **3rd Answer**: total 3=2(partial correct)+1 (partial completeness)+0(uselessness)

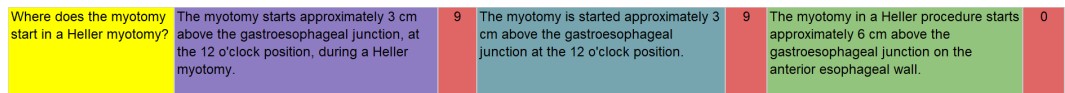

Figure 13: **1rst** : 9=6+2+1 **2nd** : 9=6+2+1 **3rd** : 0=0+0+0

## A.5 SEARCH EXAMPLES:

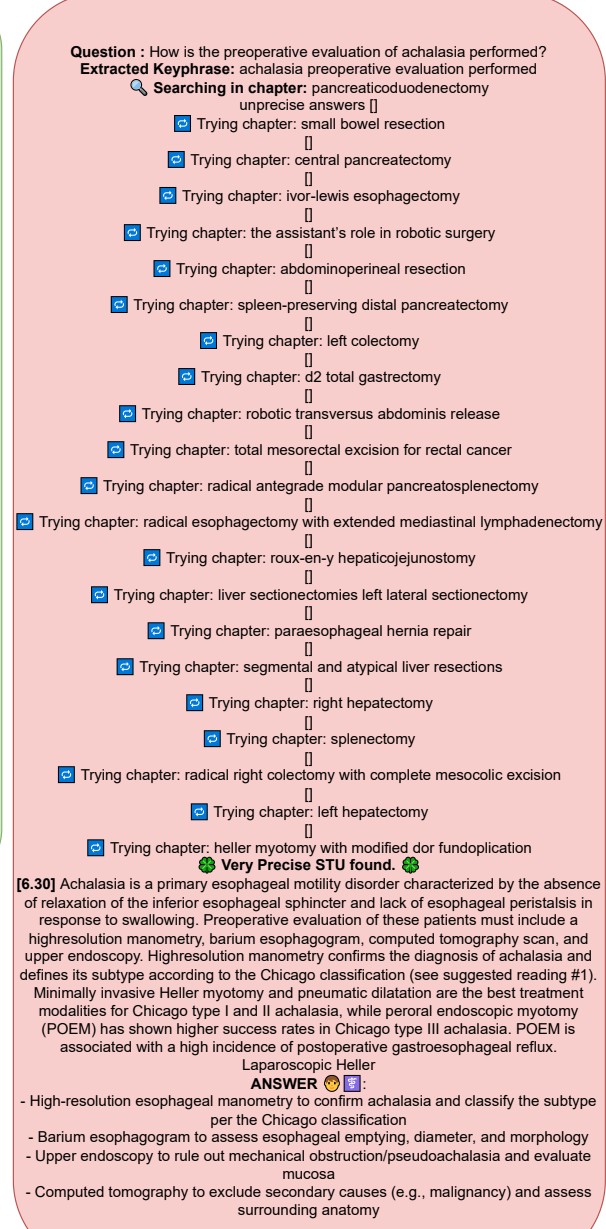

**Question:** What is the reconstruction technique performed after a robotic total gastrectomy?
**Extracted Keyphrase:** gastrectomy reconstruction robotic total technique
🔍 **Searching in chapter:** d2 total gastrectomy
🍀 **Very Precise STU found.** 🍀
--- Top Answers ---
**[score 5.78]** Intracorporeal antecolic RouxenY esophagojejunostomy using linear staplers is routinely performed after robotic total gastrectomy. After the jejunum is brought up to the transected distal esophagus in a loop fashion, an entry hole is created in the antimesenteric border of the expected anastomosis point of the jejunum. This point is 15–20 cm distal to the Treitz ligament, where no tension is present. A 45mm linear stapler can then be inserted into the holes using the R2 to create a sidetoside esophagojejunostomy. The common entry hole is subsequently closed with a 45mm stapler or by suturing. The afferent loop of the jejunum is then transected using a linear stapler. A jejunojejunostomy is finally created 45–60 cm distal to the esophagojejunostomy using a linear stapler. The entry hole is also
**ANSWER** 😀 📋:
Intracorporeal antecolic Roux-en-Y reconstruction with a linear-stapled, side-to-side esophagojejunostomy, followed by division of the afferent limb and a stapled jejunojejunostomy 45–60 cm distal.

Figure 14: Two search examples are shown, left (case 4 in Figure1), where a very precise STU is found so searching stops and the correct answer can be given immediately; and right (case 1 in Figure1), where no precise answers are found, so the system iterates through the remaining chapters (Figure 1).

**Question:** What helps to minimize parenchymal injury during traction in living donor hepatectomy?
**Extracted Keyphrase:** hepatectomy traction donor injury parenchymal living

**Searching in chapter:** living donor hepatectomy
unprecise answers [
"... Using a sponge with the ProGrasp™ in R3 can help to avoid parenchymal injury during traction...."]]

🔁 Trying chapter: roux-en-y hepaticojejunostomy []
🔁 Trying chapter: right hepatectomy
[[ "... asymmetric cutting technique ... low CVP ..."],
["... key aspect is to 'superficialize' the section line ..."]]
🔁 Trying chapter: left hepatectomy []
🔁 Trying chapter: right adrenalectomy []
🔁 Trying chapter: left adrenalectomy []
🔁 Trying chapter: small bowel resection []
🔁 Trying chapter: renal aneurysm []
🔁 Trying chapter: central pancreatectomy []
🔁 Trying chapter: left colectomy []
🔁 Trying chapter: pancreaticoduodenectomy []
🔁 Trying chapter: spleen-preserving distal pancreatectomy []
🔁 Trying chapter: the assistant's role in robotic surgery []
🔁 Trying chapter: total mesorectal excision for rectal cancer []
🔁 Trying chapter: ivor-lewis esophagectomy []
🔁 Trying chapter: bile duct injuries repair []
🔁 Trying chapter: segmental and atypical liver resections []
🔁 Trying chapter: liver sectionectomies  left lateral sectionectomy []
🔁 Trying chapter: lung upper lobectomies []
🔁 Trying chapter: kidney transplant []
🔁 Trying chapter: lung lower lobectomies []
[[ "… OR setup … stapler inserted through utility incision …" ]]
🔁 Trying chapter: enucleation of pancreatic tumors []
🔁 Trying chapter: d2 total gastrectomy []
🔁 Trying chapter: abdominoperineal resection []
🔁 Trying chapter: radical right colectomy with complete mesocolic excision []
🔁 Trying chapter: splenectomy []
🔁 Trying chapter: cholecystectomy []
🔁 Trying chapter: management of intrahepatic biliary stones []
🔁 Trying chapter: robotic transversus abdominis release []

⚠ Inaccurate answers found. Showing uncertain hits.

--- Top Answers ---
[5.46] Extraction incision often Pfannenstiel; GelPort® may help; pre-create site to avoid interrupting final steps.
[3.40]  "Using a sponge with the ProGrasp™ in R3 can help to avoid parenchymal injury during traction."
[3.21] "Superficialize" transection line to expose vessels and facilitate hemostasis (stay sutures, gravity).
[2.96] Mobilization steps; mark left side of right hepatic vein; gradual lifting with ProGrasp™.
[2.89] Lower lobectomy setup/tips on stapler port usage.

**RAG ANSWER 🤖📋:**
Using a sponge with the ProGrasp™ (in R3) during traction.

Figure 15: Search examples representing case (2) in Figure 1, this is the worst possible case when the answer is found but not recognized as very precise, therefore the model will search within all the possible chapters. This highlights the stubbornness parameter (in this case set as 28).

## A.6    Benchmark construction & Questions analysis

The benchmark was built by a three-tier committee: (i) a QA development group, (ii) a content-validation group, and (iii) an independent evaluation group. The QA development group consisted of seven clinicians (five surgeons, two medical doctors) who based their work on Giulianotti et al. (2023).

For each procedure, the QA group identified key preoperative and intraoperative knowledge and generated 1–2 questions per major textbook component (indications/contraindications, preoperative preparation, operative steps, and key risk points with tips and tricks). These were phrased as targeted queries such as "What are the indications/contraindications...?", "What precautions should be taken/avoided?", and "What is the main/key/recommended...?".

Answers were drafted by the QA group using their expert judgment to synthesize concise, clinically meaningful "standard answers" rather than copying the textbook. The content-validation group then checked each answer against the textbook, jointly reviewed for completeness, resolved discrepancies by consensus, and adopted the agreed version as ground truth.

We ran a post-hoc analysis of the full 305-question surgeon benchmark. We used simple rule-based classifiers over the question text (lexical patterns and medically grounded keywords) to assign each item to (i) a structure class (structured/factual vs. open-ended/reasoning), (ii) a Bloom-level cognitive complexity, and (iii) a reasoning type (factual recall, procedural, causal, risk, decision-making, clinical judgment).

The analysis shows that 177/305 questions (58.0%) are structured/factual and 128/305 (42.0%) are open-ended or reasoning-oriented. In terms of cognitive load, 68/305 (22.3%) fall into higher-order levels (Analyze/Evaluate/Create), with the remainder in Remember/Understand/Apply. Importantly, 39/305 (12.8%) explicitly involve decision-making, risk prediction, or clinical judgment, including 10 decision-making (3.3%), 13 risk-assessment (4.3%), 14 clinical-judgment (4.6%), and 2 clinical-evaluation (0.7%) items. The reasoning-type distribution includes factual recall (182, 59.7%), procedural knowledge (44, 14.4%), and causal reasoning (36, 11.8%), plus the decision-making and risk-related categories above. Overall, this indicates that, while factual questions are present, the benchmark also contains a substantial proportion of open-ended, reasoning-intensive items, including questions on decision-making and surgical risk assessment.

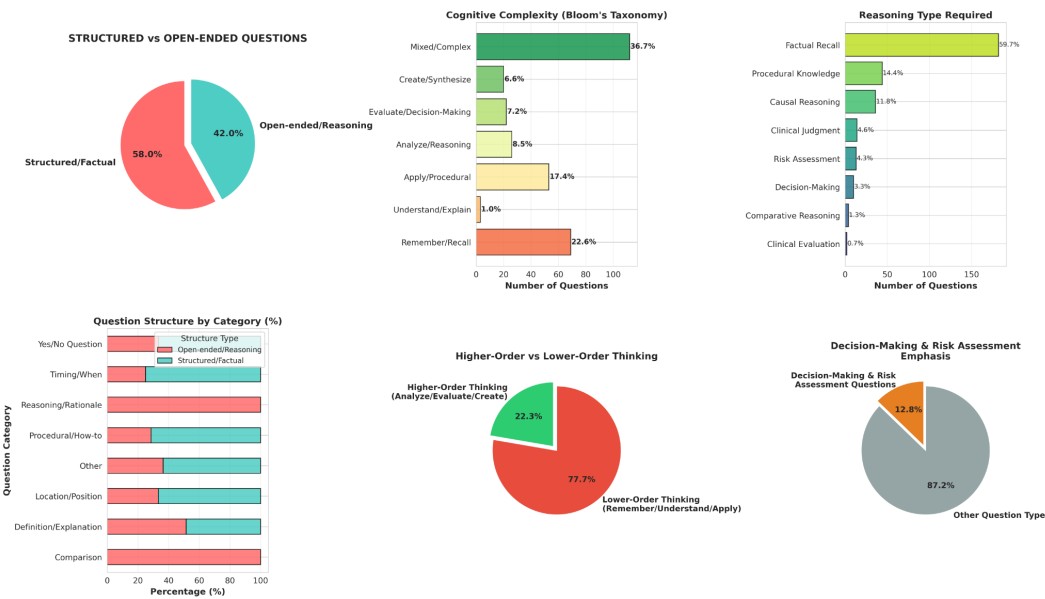

Figure 16: Distribution of question types showing that most exam items are structured/factual and target lower-order thinking (remember/understand/apply), with recall and procedural knowledge dominating. Only a smaller proportion are open-ended, higher-order questions that require analysis, evaluation, decision-making, or risk assessment.

## A.7 SMALL TEXT UNITS DISTRIBUTION

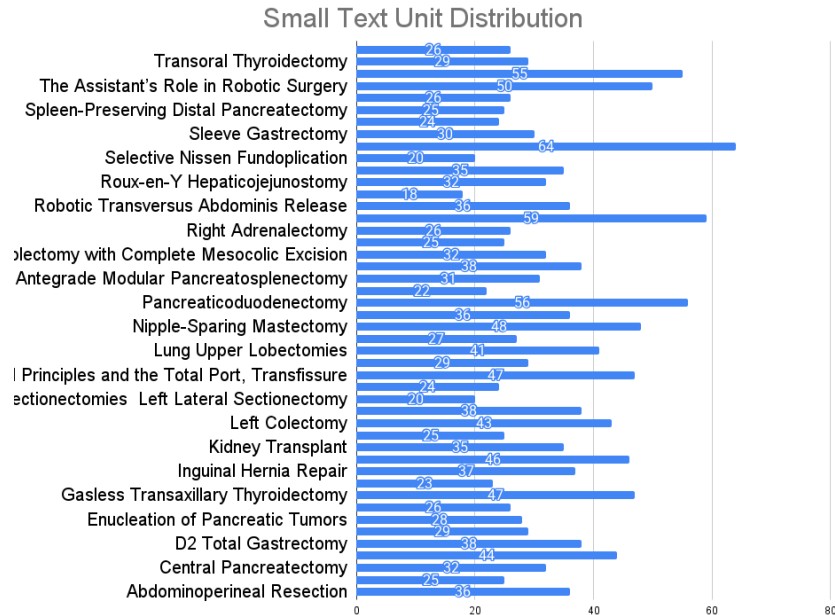

Figure 17: Histogram of STUs distribution per procedure (all the distributions are shown, but only a subset of the labels are visible for clarity)

## B REPRODUCIBILITY

All the work mentioned in this study can be found and reproduced in the following repository [1].

## C LLM USAGE

Portions of the manuscript were copy-edited using GPT-5 and Gemini 2.5 for grammar and style only. These tools were not used to draft content, perform analyses, generate data or figures, or select references. No confidential or identifiable data were provided to these services. All edits were reviewed by the authors, who take full responsibility for the final text.

---

[1]Anonymous repository: https://anonymous.4open.science/r/ICLR_2025-C1DE

