# OpenReview forum: "RASRAG: A DOMAIN-SPECIFIC RAG FRAMEWORK AND BENCHMARK FOR ROBOTIC-ASSISTED SURGERY"
_ICLR.cc/2026/Conference — Submitted to ICLR 2026_

### Official Review · Reviewer_VAre · 2025-10-27

**Soundness:** 3
**Presentation:** 3
**Contribution:** 2
**Rating:** 4
**Confidence:** 4

**Summary:**

This paper presents RASRAG, a novel domain-specific RAG framework designed to address the knowledge acquisition barriers in Robot-Assisted Surgery. Unlike traditional vector retrieval methods, RASRAG constructs authoritative RAS textbooks into a hierarchical knowledge tree and employs RankLLaMA for exploration and reranking. The work contributes the first publicly available RAS Q&A benchmark curated by a surgical team and establishes a rigorous clinical evaluation protocol. Experimental results demonstrate that RASRAG significantly outperforms conventional RAG approaches, state-of-the-art LLMs, and human expert responses across both automated metrics and blinded evaluations by independent surgeons.

**Strengths:**

- The paper introduces a novel tree-based RAG architecture that employs RankLLaMA for agentic exploration and reranking within a hierarchical knowledge structure.
- The work creates the first question-answering benchmark for RAS. This benchmark is curated by a seven-member clinical expert team including five surgeons.
- RASRAG demonstrates retrieval effectiveness largely independent of model size. Even smaller parameter models (e.g., Qwen2.5-1.5B) in RASRAG achieve retrieval performance comparable to larger models.

**Weaknesses:**

- The benchmark represents a core contribution, yet the paper inadequately describes its construction process. Key methodological details are absent, including: question generation procedures ; sources of "standard answers" (expert-authored vs. textbook excerpts?); and quality control and validation workflows.
- The core evaluation is conducted on a benchmark derived from the same textbook used to construct the knowledge base. While the paper includes a small-scale test on a second textbook, the framework's robustness across broader knowledge bases remains insufficiently validated.
- The work is highly specialized in robot-assisted surgery. The core architecture relies on a hierarchical tree structure derived from a specific textbook. Despite authors' claims of methodological generalizability, the paper provides no evidence of scalability or broader applicability.

**Questions:**

See Weakness

**Details Of Ethics Concerns:**

No Ethics Concerns.

---

> ### Author Response · Authors · 2025-11-17
>
> **Question 1**
>
> We thank the reviewer for noting that the benchmark construction needs clearer documentation. In the revised version, we explain that the dataset was built by a three-tier committee: (i) a QA development group, (ii) a content-validation group, and (iii) an independent evaluation group. The QA development group consisted of seven clinicians (five surgeons, two medical doctors) who based their work on The Foundation and Art of Robotic Surgery by Giulianotti et al. (2023).
>
> For each procedure, the QA group identified key preoperative and intraoperative knowledge and generated 1–2 questions per major textbook component (indications/contraindications, preoperative preparation, operative steps, and key risk points with tips and tricks). These were phrased as targeted queries such as “What are the indications/contraindications…?”, “What precautions should be taken/avoided?”, and “What is the main/key/recommended…?”.
>
> A quantitative analysis of the final benchmark shows that queries have diverse surface forms (e.g., 57.0% starting with “What”, 17.4% with “How”, and the remainder spread across “Which”, “Why”, “Where”, “When”, etc.), span multiple purposes (about 23.0% definition/explanation, 18.4% procedural/how-to, 5.6% explicit reasoning/rationale, plus location, timing, yes/no, and comparison questions), and cover a broad range of medical topics (general clinical considerations, technique/procedure, positioning/placement, rationale/purpose, instruments/equipment, indications, complications/risks, anatomy, measurements, contraindications, and diagnosis/evaluation). In terms of cognitive demand, 22.3% of questions require higher-order thinking (analyze/evaluate/create), 36.7% are mixed/complex, and 42.0% are classified as open-ended and reasoning-intensive (vs. 58.0% more structured/factual), with 12.8% explicitly involving decision-making or risk assessment.
>
> Answers were drafted by the QA group using their expert judgment to synthesize concise, clinically meaningful “standard answers” rather than copying the textbook. The content-validation group then checked each answer against the textbook, jointly reviewed for completeness, resolved discrepancies by consensus, and adopted the agreed version as ground truth.
>
> ---
>
> **Question 2**
>
> We agree that testing robustness on broader, more heterogeneous knowledge bases is important, but a full multi-corpus study is beyond the scope of this work, which focuses on a realistic setting where a single authoritative textbook is the primary structured source. Conceptually, RASRAG is not tied to one book: it operates on a forest-of-knowledge graph over procedures/chapters/sections. When new sources are added, (i) if they overlap (e.g., describe the same procedure), their chapters can be embedded, aligned, and merged into shared procedure nodes with source-specific child sections; (ii) if they are disjoint, they can be attached as new branches or trees, while the same hierarchical retrieval and reranking mechanism applies unchanged. Our data structures and pipeline are explicitly designed to support these multi-source extensions, and the second-textbook experiment in the paper is a first concrete step in this direction.
>
> ---
>
> **Question 3**
> We agree that RASRAG benefits from domains where knowledge is organized hierarchically and explicitly leverages this structure during retrieval. Many high-stakes domains (e.g., medicine, emergency management, environmental sciences) and common ones (e.g., do-it-yourself guides, cookbooks) naturally adopt such structured resources, making multi-step retrieval with rerankers, as in RASRAG, a justified choice to improve answer quality.
>
> To demonstrate generalization beyond the primary RAS textbook, we evaluated RASRAG on three additional textbooks: a cookbook (highly structured), an invasive plant ecology guide (structured with complex semantics), and a citizen emergency preparedness manual (simpler, with more open-ended QAs). For each book, we generated QA pairs using the same protocol adopted for the secondary RAS textbook and re-ran our experiments, showing that RASRAG is not limited to a single textbook or to RAS alone, but is applicable to any comparably hierarchical source.
>
> We are happy to evaluate RASRAG on additional books suggested by the reviewer and include those results in the final version. These tests use minimal preprocessing of the corpus (graph representation only) and a relatively small model, highlighting that the gains come from the framework rather than model scale.
>
> |Category|Model|Trial 1|Trial 2|Trial 3|Trial 4|Trial 5|Mean|Std. Dev.|
> |-|-|-:|-:|-:|-:|-:|-:|-:|
> |Plants|RASRAG+Mistral-7B|0.8000|0.8050|0.8000| 0.8050| 0.8050|0.8030|0.0024|
> |Safety|RASRAG+Mistral-7B|0.8150|0.8150|0.8150| 0.8150  | 0.8150  | 0.8150 |0.0000|
> |Cookbook|RASRAG+Mistral-7B|0.8450|0.8450|0.8450|0.8450|0.8450 |0.8450|0.0000|
>
> Refs. available in Reviewer tVGw rebuttal

---

### Official Review · Reviewer_f9dY · 2025-10-27

**Soundness:** 3
**Presentation:** 1
**Contribution:** 2
**Rating:** 4
**Confidence:** 3

**Summary:**

This paper introduces RASRAG, a domain-specific Retrieval-Augmented Generation (RAG) framework designed for Robotic-Assisted Surgery (RAS). It integrates a tree-structured retrieval system based on a surgical textbook and leverages RankLLaMA for semantic reranking at each node. The framework aims to enhance precision and contextual accuracy in surgical knowledge retrieval. The authors also propose the first RAS-specific QA benchmark, curated by surgeons and physicians, with over 300 questions reflecting real-world clinical scenarios. Extensive evaluations—including RAGAS, NVIDIA Answer Accuracy, and expert surgeon grading—demonstrate that RASRAG outperforms traditional RAG methods, fine-tuned models, and general-purpose LLMs in both factual accuracy and clinical relevance.

**Strengths:**

**Novel Domain-Specific Architecture**:
The paper effectively adapts the RAG framework to a highly specialized medical context, introducing a hierarchical “tree-of-knowledge” retrieval system that mirrors clinical reasoning structures. This design improves interpretability and retrieval precision.

**High-Quality Benchmark Creation**:
The curated QA dataset, built by surgeons and doctors, is a major contribution. It provides a reliable foundation for evaluating medical QA systems, addressing the lack of standardized evaluation in RAS.

**Comprehensive Evaluation**:
The authors conduct a multi-angle evaluation using both automated and human assessments. The inclusion of surgeon-based grading lends strong credibility and clinical grounding to the results.

**Strong Empirical Performance**:
Across all metrics, RASRAG consistently outperforms both open and proprietary baselines (e.g., GPT-4o, GPT-5), demonstrating that architecture and retrieval quality can rival model scale.

**Weaknesses:**

**Poor presentation**:
The reviewer feels uncomfortable that there is no Introduction section at the beginning and conjectures that the presentation quality seems to be quite below the expectations of the ICLR conference.

**Limited Generalization Beyond Textbook Sources**:
The framework heavily depends on a single structured textbook as its knowledge base. While a second book test is mentioned, broader generalization to heterogeneous or unstructured data (e.g., surgical notes, videos) is not explored.

**Computational Overhead**:
The use of RankLLaMA for multi-stage reranking introduces a latency of ~15 seconds per query. Although acceptable for research, this may hinder real-time clinical deployment.

**Lack of Comparison with More Recent Agentic or Planning-Based RAGs**:
The study does not deeply compare against modern multi-hop or agentic retrieval approaches (e.g., Tree-of-Thought RAG, planner-verifier pipelines), which could contextualize RASRAG’s innovation more sharply.

**Evaluation Bias Toward Structured QA**:
The benchmark and evaluations focus on structured factual questions. Open-ended, reasoning-intensive queries (e.g., decision-making or surgical risk prediction) remain underrepresented.

**Questions:**

**Scalability and Adaptation**:
How does RASRAG handle updates or integration of new medical knowledge, such as new surgical techniques or guidelines? Would retraining or structural expansion be required?

**Multimodal Extension**:
Since RAS inherently involves visual data (e.g., endoscopic imagery), could this hierarchical retrieval method be extended to incorporate multimodal (text + image/video) sources?

**Clinical Validation Path**:
Beyond expert grading, are there plans to test RASRAG’s utility in real surgical training or decision-support settings, potentially measuring time saved or error reduction?

**Model Transparency and Trust**:
Given that the framework emphasizes “traceability,” how effectively does RASRAG allow surgeons to verify retrieved evidence? Could future versions integrate explainable retrieval pathways?

---

> ### Author Response · Authors · 2025-11-17
>
> **Weakness 1**
>
> We thank the reviewer for their feedback on the presentation and apologize for any confusion caused by our current section naming. We will explicitly label as “Introduction” the text immediately following the abstract, which is designed to serve this function.
>
> ---
>
> **Weakness 2**
>
> To demonstrate generalizability, we tested our approach on three additional textbooks. Due to space constraints, we cannot provide more detail here and apologize; please see reviewer tVGw’s Question 1 rebuttal for full details.
>
> ---
>
> **Weakness 3**
>
> We acknowledge the concern regarding the ~15 seconds latency introduced by RankLLaMA-based multi-stage reranking. In the current implementation, RASRAG is designed for high-quality, traceable reasoning rather than real-time reactive control. For context, the per-question latencies we report for other RAG pipelines are substantially higher: 62.22 s for Paper QA and 48.70 s for MedRAG. Moreover, as we discuss in our response on the latency–performance trade-off, limiting the maximum number of search calls and tuning the $k_{\max}$ exploration cap allows us to control latency while preserving most of the quality gains.
>
> ---
>
> **Weakness 4**
>
> We systematically considered RAG frameworks that are conceptually closest to RASRAG, and we believe some of the approaches mentioned by the reviewer would broaden the comparison scope beyond what is feasible for this study.
>
> ---
>
> **Weakness 5**
>
> In light of this comment, we ran a post-hoc analysis of the full 305-question surgeon benchmark. We used simple rule-based classifiers over the question text (lexical patterns and medically grounded keywords) to assign each item to (i) a structure class (structured/factual vs. open-ended/reasoning), (ii) a Bloom-level cognitive complexity, and (iii) a reasoning type (factual recall, procedural, causal, risk, decision-making, clinical judgment).
>
> The analysis shows that 177/305 questions (58.0%) are structured/factual and 128/305 (42.0%) are open-ended or reasoning-oriented. In terms of cognitive load, 68/305 (22.3%) fall into higher-order levels (Analyze/Evaluate/Create), with the remainder in Remember/Understand/Apply. Importantly, 39/305 (12.8%) explicitly involve decision-making, risk prediction, or clinical judgment, including 10 decision-making (3.3%), 13 risk-assessment (4.3%), 14 clinical-judgment (4.6%), and 2 clinical-evaluation (0.7%) items. The reasoning-type distribution includes factual recall (182, 59.7%), procedural knowledge (44, 14.4%), and causal reasoning (36, 11.8%), plus the decision-making and risk-related categories above. Overall, this indicates that, while factual questions are present, the benchmark also contains a substantial proportion of open-ended, reasoning-intensive items, including questions on decision-making and surgical risk assessment.
>
> **Question 1**
>
> RASRAG does not require training as it is an RAG architecture. Specializing a reranker to a specific domain would be very interesting future work and is a possible high advantage for our architecture. New knowledge would just need to be incorporated into the context database in an additive way (creating a new node in the graph). This is straightforward in the current implementation, as the code uses a nested dictionary.
>
> ---
>
> **Question 2**
>
> With the packages we use to treat the PDF file (fitz), it is possible to extract all images and hyperlinks (for example, the book we used for the test contains a lot of hyperlinks directly to video examples). Then, given a chunk of text, providing the correct answer to the query is simply a string association to extract the correct stored image or to direct to the correct hyperlink. We tested this and, if necessary, can show an example. We decided not to include that, as technically, there is no innovation, and this is not where we believe RASRAG shone the most.
>
> ---
>
> **Question 3**
>
> RASRAG will primarily be used as an online copilot paired with a foundation model for video interpretation, acting as a slower, specialized, high-level planner for robotic-assisted surgery. Therefore, we do not consider an average latency of 15 seconds per query to be a significant issue. In practice, procedures progress much more slowly at the level of granularity we target (goal planning rather than low-level action planning).
>
> ---
>
> **Question 4**
>
> The current version can exactly pinpoint where the information is coming from in the database. We would like the reviewer to have a look at Appendix A.5, the third example (p. 24). For example, when in the graph we have the chunk:
>
> > "... Using a sponge with the ProGrasp(TM) in R3 can help to avoid parenchymal injury during traction ..."
>
> This is associated in our code with exactly where in the database it is coming from. Future versions could further expose these retrieval paths explicitly in the user interface to strengthen model transparency and trust.

---

### Official Review · Reviewer_cNBn · 2025-10-29

**Soundness:** 3
**Presentation:** 2
**Contribution:** 2
**Rating:** 4
**Confidence:** 3

**Summary:**

This paper proposes RASRAG, a domain-specific Retrieval-Augmented Generation (RAG) framework for Robotic-Assisted Surgery (RAS). The method operates by structuring a source RAS textbook into a hierarchical knowledge. It then employs a RankLLaMA-based model to perform semantic reranking and navigation through this hierarchy, moving from high-level procedures (BTUs) down to specific text chunks (STUs) . A key contribution is the introduction of a new, 305-pair question-answer (QA) benchmark curated by a team of seven clinicians. The framework's performance is assessed using automated metrics (RAGAS, NVIDIA Answer Accuracy) and a blind evaluation conducted by three independent surgeons.

**Strengths:**

- The paper focuses on a clear and important real-world problem domain: Robotic-Assisted Surgery (RAS). This field faces distinct challenges, including a shortage of trained surgeons , barriers to training, and limited access to specialized academic materials.
- The creation and release of first-of-its-kind QA benchmark curated by clinical experts (surgeons and physicians) is a valuable contribution, providing a new resource for future research in this area.
- The explanation of the current status of RAS and its critical challenges is reasonable and supports the understanding of the RAS environment and the intent of the framework.
- The evaluation design is comprehensive, incorporating automated RAG metrics, clinical answer accuracy metrics, and a blind human expert evaluation, which represents a robust approach to validation.

**Weaknesses:**

### Limited Methodological Novelty and Mismatch with Domain
The core method, a hierarchical search through a structured corpus is fundamentally just a structured search over a single textbook's table of contents. The validation for this (Appendix A.1) demonstrates properties of a well-organized textbook, not unique properties of the *RAS domain*. The authors themselves concede the method's generality ("This methodology could generalize well beyond RAS"), which undermines the central claim of domain-specific innovation.

### Under-described Benchmark
A primary contribution, the 305-pair QA benchmark, is presented with insufficient detail. Section 2 merely states *who* created it (7 clinicians) and *what* it is (305 pairs). Critical information regarding the protocol for question generation, the quality assurance process, answer curation process. This lack of transparency makes it difficult to assess the benchmark's quality or reproducibility.

### Worries of overfitting to the retrieval corpus
The framework relies on heuristic, rule-based procedures (e.g., select definite and candidate passages). Because the approach was tuned to the specific retrieval corpus, its evaluation primarily demonstrates properties of that corpus’s organization (and the chosen search heuristics) rather than unique features of the RAS domain. The resulting complexity might introduces extra engineering burden. While the complexity of the RAS domain is understandable, the heuristic-based approach may limit the method’s extensibility; therefore, the paper should provide a stronger justification to address this concern.

### Needs more baseline
The authors rely on a latency-heavy tree-search structure to achieve gains, but do not sufficiently evaluate stronger or refined similarity-based baselines (e.g., Qwen-based retriever, re-ranking with lightweight cross-encoders, iterative RAG loops, or search-agent framework for complicate queries). Though the authors includes a few variations in table 2 (MedGraph, PaperQA), I wonder whether it is sufficiently represent the potential of existing RAG researches (and also think it should be included in table 1.)

**Questions:**

### Chapter-level independence as a specific property of RAS
The "Chapter-level conditional independence" seems to be the main justification for the hierarchical tree structure. Can you elaborate on why this is a specific property of RAS knowledge, rather than a general property of any well-structured textbook? How would this method perform on a non-hierarchical corpus, such as 10,000 individual surgical case reports?

### Regarding the benchmark
What was the detailed protocol given to the 7 clinicians for generating questions and answers? What quality control measures were in place to ensure the answers were correct, consistent, and comprehensive before using them as ground truth?

### Correlation with general performance of models
In table 1, unlike my expectation, large-scale models (including close-sourced) have relatively lower performance (context precision, context recall) tendency than smaller open-source models. It would be helpful for me to get a justification of this.

### Latency-Performance Tradeoff
The work needs a more rigorous, quantitative comparison that measures both quality gains and latency/compute costs. Could the authors limit the number of search call (or iterations) and evaluate the performance?



### paper error
In table 1, the performance of context precision is wrongly highlighted. the best performance is Qwen2.5-1.5B-Instruct (0.8918), not MedGemma (0.8829).

---

> ### Author Response · Authors · 2025-11-18
>
> **Question 1**
> We thank the reviewer for this insightful question, which helps clarify why we adopt a hierarchical approach in the robot-assisted surgery (RAS) domain. While “chapter-level conditional independence” could in principle hold for any well-structured textbook, our aim is to tailor the method to RAS. RAS textbooks exhibit an unusually strong hierarchy: chapters almost always correspond to distinct, complete surgical procedures that may share actions (e.g., dissection, suturing) but are, surgically, semantically unique and independent due to anatomical specificity and procedure-specific methodology. The “chapter-level independence” assumption formalizes this key property. We argue that the specialized, high-stakes nature of RAS justifies a dedicated AI approach, distinct from generalized methods for broader domains (e.g., Activities of Daily Living).
>
> For a non-hierarchical corpus such as 10 000 surgical case reports, we would expect RASRAG to be less suitable, particularly in execution time rather than precision. Such a corpus reflects wide variation in surgeon skills, preferences, and valid technique variants, so each query would have to search many heterogeneous instances, and the notion of a single “best” answer is less well-defined. We therefore view RASRAG as intentionally specialized for structured, procedure-centric knowledge sources such as RAS textbooks.
>
> ---
>  **Question 2**
>
> We thank the reviewer for this insightful comment and refer them to our response to Reviewer VAre, Question 1. We apologize for not elaborating further here, as this rebuttal is already heavily constrained by the character limit.
>
> ---
>
>  **Question 3**
> Our analysis suggests that this counterintuitive trend is largely an artifact of model behavior and how RAGAS metrics are computed, and in revisiting the evaluation we also identified a small discrepancy between the scripts used for closed models (GPT-5, Gemini-2.5 Pro) and those for open-weight models, which caused RASRAG sometimes (~10 times) to retrieve one  additional contexts and thus reduced context accuracy. We are rerunning this evaluation with the corrected setup and will update the corresponding rows in the rebuttal in the coming days. Thank you for pointing this out.
>
> However, proprietary large-scale models tend to generate fluent, human-like paraphrases of the RASRAG context, whereas many smaller open-source models often copy it verbatim. Because automated metrics such as RAGAS, especially faithfulness, depend heavily on textual overlap between answer and context, a large model that correctly paraphrases may receive a lower score than a smaller model that simply reproduces the source wording. This discrepancy likely stems from stylistic effects on the scoring mechanism rather than weaker reasoning by larger models. We therefore suggest interpreting RAGAS metrics together with NVIDIA metrics (Table 2) for a fuller view of answer quality.
>
> Below, we show an example where a correct paraphrased answer is penalized by RAGAS:
>
> **Example QA and RAGAS scores**
>
> |Field|Value|
> |-|-|
> |**user_input**|What is one of the reasons for dysphagia after a Nissen fundoplication?|
> |**retrieved contexts**|[...] Nissen [...], it is not immune from side effects and complications, such as dysphagia[...]. One of the main reasons for dysphagia is a tight wrap. [...].|
> |**response**|A tight wrap, where the fundus of the stomach is wrapped too tightly around the esophagus, is one of the main reasons for dysphagia (difficulty swallowing) after a Nissen fundoplication.|
> |**GT**|One of the main reasons for dysphagia is a tight wrap.|
>
> ||context precision|context recall|faithfulness|answer relevancy|semantic similarity|
> |-|-|-|-|-|-|
> |**score**|1.00|1.00|**0.33**|1.00|0.84|
>
> ---
>
>  **Question 4**
> We agree that a quantitative analysis of the quality–latency trade-off is important. Following this suggestion, we are running additional experiments (reported in Appendix A.2) that tune key hyperparameters. Below is the first result, including limiting the maximum number of search calls ($k_{\max}$). This has been conducted on 60 randomly sampled questions using RASRAG+Llama-3.2-1B-Instruct:
>
> |$k_{max}$|T(s)|NvA|‖|$k_{max}$|T(s)|NvA|
> |-|-|-|-|-|-|-|
> |0|8.32|0.6932|‖|15|11.86|0.7315|
> |1|9.27|0.7041|‖|16|11.96|0.7269|
> |2|9.45|0.7000|‖|17|12.12|0.7130|
> |3|9.70|0.6961|‖|18|12.22|0.7500|
> |4|9.93|0.7500|‖|19|12.33|0.7222|
> |5|10.03|0.7452|‖|20|12.44|0.7361|
> |6|10.39|0.7358|‖|21|13.32|0.7591|
> |7|10.53|0.7075|‖|22|13.42|0.7727|
> |8|10.65|0.6840|‖|23|13.52|0.7182|
> |9|10.80|0.7500|‖|24|13.92|0.7364|
> |10|10.87|0.7123|‖|25|13.73|0.7273|
> |11|11.36|0.7315|‖|26|13.86|0.7182|
> |12|11.50|0.6991|‖|27|13.95|0.7136|
> |13|11.69|0.7037|‖|28|14.07|0.7227|
> |14|11.76|0.6991|‖|29|14.10|0.7273|
>
> Latency trend: slope = 0.1851 sec per $k_{max}$ unit;
> Nvidia accuracy trend: slope = 0.000798 per  $k_{max}$ unit
>
> This will be added to the other quality–latency analyses, which are reported in Appendix A.2.

---

> > ### Author Response · Authors · 2025-11-25
> >
> > **Question 3 – Continuation**
> >
> > After running the evaluation again with the corrected setup, the updated aggregated scores for RASRAG are:
> >
> > | Model          | Context precision | Context recall | Faithfulness | Answer relevancy | Semantic similarity |
> > |----------------|-------------------|----------------|-------------|------------------|---------------------|
> > | Gemini-2.5-Pro | 0.8608            | 0.8334         | 0.8300      | 0.8138           | 0.7766              |
> > | GPT-5          | 0.8502            | 0.8202         | 0.7685      | 0.6485           | 0.7631              |
> >
> > We observe some instability in the scores, particularly for context precision. In cases where an answer is supported by numerous retrieved contexts, the RAGAS framework sometimes assigns a precision score of 0 even though at least one of the retrieved contexts is correct.

---

### Official Review · Reviewer_tVGw · 2025-11-01

**Soundness:** 2
**Presentation:** 2
**Contribution:** 2
**Rating:** 4
**Confidence:** 4

**Summary:**

RASRAG is introduced as a domain-specialized retrieval-augmented generation framework aimed at improving medical report generation and question-answering in a specific clinical domain. The method builds a hierarchical “forest of knowledge” from a key domain textbook, allowing an LLM agent to iteratively explore and rerank relevant sections, much like an expert searching through a textbook.
The authors also contribute a new expert-curated benchmark of question–answer pairs reflecting real clinical queries, along with an evaluation protocol.

**Strengths:**

The paper tackles a well-identified gap by focusing on a specialized medical domain where general LLMs underperform. The motivation is explained with real-world context (e.g. limited access to expert knowledge in the domain), making the case that a domain-specific model is needed and valuable.
The paper contributes a new expert-curated QA benchmark for the domain, which is a valuable resource for the community.

**Weaknesses:**

1) The approach is tailored to a specific domain and relies on a structured hierarchy for retrieval. This dependence means that applying RASRAG to a different domain would require a similarly well-structured knowledge source. If the domain knowledge is not organized as this, the performance may degrade.
2) While the results are strong, the paper could benefit from deeper ablation studies or analysis of each component in the pipeline.
3) The custom QA benchmark, while valuable, is relatively small in scale (on the order of a few hundred expert-curated questions). This raises a concern that the evaluation, though high quality, might not cover the full diversity of real-world queries.

**Questions:**

please address the concerns in weakness sections.

---

> ### Author Response · Authors · 2025-11-17
>
> **Question 1**
>
> We agree that RASRAG benefits from domains where knowledge is organized hierarchically and explicitly leverages this structure during retrieval. Many high-stakes domains (e.g., medicine, emergency management, environmental sciences) and common ones (e.g., do-it-yourself guides, cookbooks) naturally adopt such structured resources, making multi-step retrieval with rerankers, as in RASRAG, a justified choice to improve answer quality.
>
> To demonstrate generalization beyond the primary RAS textbook, we evaluated RASRAG on three additional textbooks: a cookbook (highly structured), an invasive plant ecology guide (structured with complex semantics), and a citizen emergency preparedness manual (simpler, with more open-ended QAs). For each book, we generated QA pairs using the same protocol adopted for the secondary RAS textbook and re-ran our experiments, showing that RASRAG is not limited to a single textbook or to RAS alone, but is applicable to any comparably hierarchical source.
>
> We are happy to evaluate RASRAG on additional books suggested by the reviewer and include those results in the final version. These tests use minimal preprocessing of the corpus (graph representation only) and a relatively small model, highlighting that the gains come from the framework rather than model scale.
>
>
>
> |Category|Model|Trial 1|Trial 2|Trial 3|Trial 4|Trial 5|Mean|Std. Dev.|
> |-|-|-:|-:|-:|-:|-:|-:|-:|
> |Plants|RASRAG+Mistral-7B|0.8000|0.8050|0.8000|0.8050|0.8050|0.8030|0.0024|
> |Safety|RASRAG+Mistral-7B|0.8150|0.8150|0.8150|0.8150|0.8150|0.8150|0.0000|
> |Cookbook|RASRAG+Mistral-7B|0.8450|0.8450|0.8450|0.8450|0.8450|0.8450|0.0000|
>
>
>
> **References (all available for free online):**
>
> 1. National Institutes of Health, Health and Human Services Department. *Deliciously Healthy Dinners*. 2008.
> 2. Huebner, C. D., and Jones, T. *Invasive Plants Field and Reference Guide: An Ecological Perspective of Plant Invaders of Forests & Woodlands*. US Forest Service, 2022.
> 3. Federal Emergency Management Agency. *Are You Ready?: An In-Depth Guide to Citizen Preparedness*. FEMA, 2013.
>
>
> ---
>
> **Question 2**
>
> We appreciate the reviewer's suggestion to deepen the analysis of our pipeline components. Our paper already includes several key comparisons, such as the performance of different retrieval methods (e.g., conventional cosine similarity vs. state-of-the-art RAG methods) and the significant impact of using RASRAG versus not using it, particularly with state-of-the-art proprietary models.
>
> To further enhance this analysis, we are expanding Appendix A.2. As noted in our response regarding the latency–performance trade-off, this updated section will provide a more detailed analysis of hyperparameter tuning, including the k_max exploration cap, which was also pointed out by another reviewer. This new analysis will serve as a deeper ablation study on key components of our retrieval pipeline, directly addressing the trade-offs between performance and computational cost. We will report these results in this forum as soon as we have them. If the reviewer has any particular ablation study in mind, we would be happy to try to do it.
>
> ---
>
> **Question 3**
>
> We acknowledge that our expert-curated QA benchmark is smaller than web-scale datasets and thus may not capture the full diversity of all possible real-world queries. Its size reflects the fact that each question and reference answer is manually authored and refined by clinically active surgeons highly specialized in robotic-assisted surgery. Their expertise is rare, time-consuming, and costly.
>
> We argue that the benchmark nonetheless offers high ecological validity: questions are drawn from real clinical decision-making scenarios and reflect realistic information needs. The surgeons involved span multiple countries and levels of seniority, which we believe contributes to diversity in query styles and perspectives. Due to double-blind review, we cannot disclose identifying details now, but upon acceptance, we will provide aggregated statistics on their backgrounds (e.g., geographic distribution, years of experience, subspecialty). Below we show two anonymized examples:
>
> | Surgeon | Total # procedures | Laparoscopic cases | mid-complexity surgeries |Highly complex surgeries|Training hours surgical robotics |
> |-|-|-|-|-|-|
> | A| 1500| 900| 1125| 375| ~500|
>
> |Surgeon|Cholecystectomies|Inguinal hernias|Colon procedures|Bariatric procedures|Esophageal procedures|Training hours surgical robotics |
> |-|-|-|-|-|-|-|
> |B|1000|200|100|100|50| ≥100|
>
> We are already working with the same expert panel to expand the benchmark and provide richer descriptive statistics (e.g., distributions over question types, difficulty, and topic coverage). We intend to release it as an open, evolving resource for the community and will include preliminary statistics in the revised version.

---

### Meta-Review · Area_Chair_wLqz · 2026-01-11

**Summary:**

The paper introduced a retrieval framework RASRAG (Robotic-Assisted Surgery Retrieval-Augmented Generation) designed to navigate complex surgical textbooks. In the first reviews, many concerns were raised such as the framework too dependent on structured textbook data (Reviewer tVGw & Reviewer f9dY & Reviewer VAre), limited methodology novelty (Reviewer cNBn), poor presentation & lack of agentic RAG comparisons (Reviewer f9dY), lack of details on the benchmark construction (Reviewer VAre). As a result the initial scores are all marginally below the acceptance threshold. The rebuttal seems to address some of the concerns but there are still remaining concerns such as the methodology novelty and the potential real-world clinical generalization issue regarding the textbook-centered benchmark. During the discussion session, there has been no reviewer engagement since the rebuttal was posted.

Overall, in the AC’s assessment, the paper aims to make two primary contributions: (1) Methodological contribution: A hierarchical "Tree-RAG" architecture (Forest of Knowledge) that utilizes RankLLaMA for agentic navigation through structured textbook content; and (2) A new benchmark: 305 QA pairs from textbooks in the RAS domain. However, based on the reviews and the authors’ rebuttal, these two contributions are not yet sufficiently substantiated.

While the authors have added numerous comparison experiments and additional results drawn from textbooks, these additions do not fully address the core concerns raised by the reviewers. From a methodological standpoint, the underlying search strategy (e.g., tree-based traversal) does not appear to introduce a fundamentally new concept beyond existing approaches. From a benchmarking perspective, the paper does not clearly demonstrate the real-world clinical or practical utility of the proposed benchmark, particularly given its reliance on textbook-derived data. As a result, neither contribution—methodological nor benchmarking—emerges as sufficiently compelling in its current form. The paper might benefit from focusing more deeply on one direction, as the current attempt to advance both simultaneously dilutes the overall impact.

Given the limited reviewer discussion following the rebuttal and the lack of substantive engagement beyond the initial reviews, the AC anticipates that, while reviewers ttVGw and f9dY may increase their scores to 6, the remaining two reviewers are likely to maintain their original scores of 4. This would result in an overall score of approximately 5, which falls below the acceptance threshold. Overall, the AC finds the paper interesting and potentially promising, but believes it requires further refinement. In its current form, it sits near the borderline and slightly leans toward rejection.

**Reviewer Concerns:**

The major concerns raised in the initial reviews—regarding weak presentation, limited benchmark description, lack of comparisons, and unclear generalization beyond a single textbook—have been substantially addressed in the revision. The authors have added significant details, expanded comparison experiments, and included new evaluations on additional textbooks. These revisions have definitely improved the paper’s clarity and completeness.

However, the core concern regarding methodological novelty remains insufficiently addressed. From the AC’s perspective, the central logic of the proposed framework is essentially an engineering implementation of a structured search over a Table of Contents. While this represents a reasonable and potentially useful heuristic for high-stakes domains, it does not constitute a fundamental advance in retrieval theory or large language model architecture. The proposed “Tree-RAG” approach appears to be an incremental extension of existing reranking and structured retrieval techniques applied to a curated dataset, rather than a qualitatively new method.

Regarding the benchmark, the reviewers acknowledge the value of expert curation and the care taken in dataset construction. Nevertheless, a notable gap remains between textbook-derived knowledge and real-world clinical practice. The reliance on a single authoritative source limits the benchmark’s ability to capture the inherent complexity and variability of clinical settings, including unstructured clinical notes, procedural variability, and multimodal data. Although the evaluation is thorough, it primarily reflects the organizational structure of the textbooks rather than the model’s capacity to reason over heterogeneous, noisy, and incomplete clinical information.

Overall, the AC concurs believes that the paper would be more compelling if it either introduced a fundamentally new retrieval paradigm or provided stronger real-world clinical validation beyond textbook-based excerpts.

**Reviewer Scores:**

Given that the rebuttal did not convincingly address concerns regarding methodological novelty or the benchmark’s generalization to real-world scenarios, and considering the limited reviewer engagement during the discussion period, the AC anticipates that while reviewers ttVGw and f9dY may raise their scores to 6, the remaining two reviewers are likely to maintain their original scores of 4. This would yield an overall score of approximately 5, which remains below the acceptance threshold.

Overall, the AC finds the paper interesting and potentially promising, but believes it requires further refinement. In its current form, it sits near the borderline and slightly leans toward rejection.

---

### Decision · Program_Chairs · 2026-01-26

Reject